# Evaluation and uncertainty investigation of the NO₂, CO and NH₃ modeling over China under the framework of MICS-Asia III

Lei Kong[1,3], Xiao Tang[2,3], Jiang Zhu[1,3], Zifa Wang[2,3], Joshua S. Fu[4], Xuemei Wang[5], Syuichi Itahashi[6,7], Kazuyo Yamaji[8], Tatsuya Nagashima[9], Hyo-Jung Lee[10], Cheol-Hee Kim[10], Chuan-Yao Lin[11], Lei Chen[2,3], Meigen Zhang[2,3], Zhining Tao[12,13], Jie Li[2,3], Mizuo Kajino[14,15], Hong Liao[16], Zhe Wang[17,2], Kengo Sudo[18] Yuesi Wang[2,3], Yuepeng Pan[2,3], Guiqian Tang[2,3], Meng Li[19,20], Qizhong Wu[21,22]; Baozhu Ge[2,3], Gregory R. Carmichael[23]

[1]ICCES, Institute of Atmospheric Physics, Chinese Academy of Sciences, Beijing, 100029, China
[2]LAPC, Institute of Atmospheric Physics, Chinese Academy of Sciences, Beijing, 100029, China
[3]University of Chinese Academy of Sciences, Beijing, 100049, China
[4]Department of Civil and Environmental Engineering, University of Tennessee, Knoxville, TN, 37996, USA
[5]Institite for Environment and Climate Research, Jinan University, Guangzhou, 510632, China
[6]Central Research Institute of Electric Power Industry, Abiko, Chiba, 270-1194, Japan
[7]Department of Marine, Earth, and Atmospheric Sciences, North Carolina State University, Raleigh, NC 27607, USA
[8]Graduate School of Maritime Sciences, Kobe University, Kobe, Hyogo 658-0022, Japan
[9]National Institute for Environmental Studies, Onogawa, Tsukuba 305-8506, Japan
[10]Department of Atmospheric Sciences, Pusan National University, Busan, 46241, South Korea
[11]Research Center for Environmental Changes, Academia Sinica, Taipei, 115, Taiwan
[12]Universities Space Research Association, Columbia, MD, USA
[13]NASA Goddard Space Flight Center, Greenbelt, MD, 130, USA
[14]Meteorological Research Institute, Japan Meteorological Agency, Tsukuba, Ibaraki, 305-0052, Japan
[15]Faculty of Life and Environmental Sciences, University of Tsukuba, Tsukuba, Ibaraki, 305-8577, Japan
[16]School of Environmental Science and Engineering, Nanjing University of Information Science & Technology, Nanjing 210044, China
[17]Research Institute for Applied Mechanics (RIAM), Kyushu University, Fukuoka, Japan
[18]Graduate School of Environmental Studies, Nagoya University, Nagoya, Japan
[19]Ministry of Education Key laboratory for Earth System Modeling, Department of Earth System Science, Tsinghua University, Beijing, 100084, China
[20]Multiphase Chemistry Department, Max Planck Institute for Chemistry, Mainz, 55128, Germany
[21]College of Global Change and Earth System Science, Beijing Normal University, Beijing 100875, China
[22]Joint Centre for Global Changes Studies, Beijing Normal University, Beijing 100875, China
[23]Center for Global and Regional Environmental Research, University of Iowa, Iowa City, IA, 52242, USA

*Correspondence to*: Xiao Tang(tangxiao@mail.iap.ac.cn)

**Abstract.** Despite the significant progress in improving the chemical transport models (CTMs), applications of these modeling endeavours are still subject to the large and complex model uncertainty. Model Inter-Comparison Study for Asia III (MICS-Asia III) has provided the opportunity to assess the capability and uncertainty of current CTMs in East Asia applications. In this study, we have evaluated the multi-model simulations of nitrogen dioxide (NO₂), carbon monoxide (CO) and ammonia (NH₃) over China under the framework of MICS-Asia III. Thirteen modeling results, provided by several independent groups from different countries/regions, were used in this study. Most of these models used some modeling domain with a horizontal resolution of 45km, and were driven by common emission inventories and meteorological inputs. New observations over North

China Plain (NCP) and Pearl River Delta (PRD) regions were also available in MICS-Asia III, allowing the model evaluations
over highly industrialized regions. The evaluation results show that most models well captured the monthly and spatial patterns
of NO₂ concentrations in the NCP region though NO₂ levels were slightly underestimated. Relatively poor performance in
NO₂ simulations was found in the PRD region with larger root mean square error and lower spatial correlation coefficients,
which may be related to the coarse resolution or inappropriate spatial allocations of the emission inventories in the PRD region.
All models significantly underpredicted CO concentrations in both the NCP and PRD regions, with annual mean concentrations
65.4% and 61.4% underestimated by the ensemble mean. Such large underestimations suggest that CO emissions might be
underestimated in current emission inventory. In contrast to the good skills in simulating the monthly variations of NO₂ and
CO concentrations, all models failed to reproduce the observed monthly variations of NH₃ concentrations in the NCP region.
Most models mismatched the observed peak in July and showed negative correlation coefficients with observations, which
may be closely related to the uncertainty in the monthly variations of NH₃ emissions and the NH₃ gas-aerosol partitioning.
Finally, model inter-comparisons have been conducted to quantify the impacts of model uncertainty on the simulations of these
gases which are shown increase with the reactivity of species. Models contained more uncertainty in the NH₃ simulations. This
suggests that for some highly active and/or short-lived primary pollutants, like NH₃, model uncertainty can also take a great
part in the forecast uncertainty besides the emission uncertainty. Based on these results, some recommendations are made for
future studies.

# 1 Introduction

As the rapid growth in East Asia's economy with surging energy consumption and emissions, air pollution has become
an increasingly important scientific topic and political concern in East Asia due to its significant environmental and health
effects (Anenberg et al., 2010;Lelieveld et al., 2015). Chemical transport models (CTMs), serving as a critical tool in both the
scientific research and policy makings, have been applied into various air quality issues, such as air quality prediction, long-
range transport of atmospheric pollutants, development of emission control strategies and understanding of observed chemical
phenomena (e.g. Cheng et al., 2016;Li et al., 2017a;Lu et al., 2017;Ma et al., 2019;Tang et al., 2011;Xu et al., 2019;Zhang et
al., 2019). Nevertheless, air quality modeling remains a challenge due to the multi-scale and non-linear nature of the complex
atmospheric processes (Carmichael et al., 2008). It still suffers from large uncertainties related to the missing or poorly
parameterized physical and chemical processes, inaccurate and/or incomplete emission inventories as well as the poorly
represented initial and boundary conditions (Carmichael et al., 2008;Dabberdt and Miller, 2000;Fine et al., 2003;Gao et al.,
1996;Mallet and Sportisse, 2006). Understanding such uncertainties and their impacts on the air quality modeling is of great
importance in assessing the robustness of models for their applications in scientific research and operational use.
There are specific techniques to assess these uncertainties. Monte Carlo simulations, based on different values of model
parameters or input fields sampled from a predefined probability density function (PDF), can provide an approximation to the
PDF of possible model output and serves as an excellent characterization of the uncertainties in simulations (Hanna et al.,
2001). However, this method is more suited to deal with the uncertainty related to the continuous variables, such as input data
or parameters in parameterization. The ensemble method, based on a set of different models, is an alternative approach to
accounting for the range of uncertainties (Galmarini et al., 2004;Mallet and Sportisse, 2006).   For example, the Air Quality
Model Evaluation International Initiative (AQMEII) has been implemented in Europe and North America to investigate the
model uncertainties of their regional-scale model predictions (Rao et al., 2011). To assess the model performances and
uncertainties in East Asia applications, the Model Inter-Comparison Study for Asia (MICS-Asia) has been initiated in year
1998. The first Phase of MICS-Asia (MICS-Asia I) was carried out during period 1998–2002, mainly focusing on the long-
range transport and depositions of sulfur in Asia (Carmichael et al., 2002). In 2003, the second phase (MICS-Asia II) was
initiated and took more species related to the regional health and ecosystem protection into account, including nitrogen
compounds, $O_3$ and aerosols. Launched in 2010, MICS-Asia III has greatly expanded its study scope by covering three
individual and interrelated topics: (1) evaluate strength and weaknesses of current multi-scale air quality models and provide
techniques to reduce uncertainty in Asia; (2) develop a reliable anthropogenic emission inventories in Asia and understanding
uncertainty of bottom-up emission inventories in Asia; (3) provide multi-model estimates of radiative forcing and sensitivity
analysis of short-lived climate pollutants.
This study addresses one component of topic 1, focusing on the three gas pollutants of $NO_2$, CO and $NH_3$. Compared with
MICS-Asia II, more modeling results (fourteen different models with thirteen regional models and one global model) were
brought together within the topic 1 of MICS-Asia III, run by independent modeling groups from China, Japan, Korea, United
States of America and other countries/regions. The different models contain differences in their numerical approximations
(time step, chemical solver, etc.) and parameterizations, which represent a sampling of uncertainties residing in the air quality
modeling. However, it would be difficult to interpret the results from inter-comparison studies when the models were driven
by different meteorological fields and emission inventories. Thus, in MICS-Asia III the models were constrained to be operated
under the same conditions by using the common emission inventories, meteorological fields, modeling domain and horizontal
resolutions. The simulations were also extended from the four months in MICS-Asia II to one-full year of 2010.
$NO_2$, CO and $NH_3$ are three important primary gas pollutants that has wide impacts on the atmospheric chemistry. As a
major precursor of $O_3$, $NO_2$ plays an important role in the tropospheric $O_3$ chemistry, and also contributes to the rainwater
acidification and the formation of secondary aerosols (Dentener and Crutzen, 1993;Evans and Jacob, 2005). CO is a colorless
and toxic gas ubiquitous throughout the atmosphere which is of interest as an indirect greenhouse gas (Gillenwater, 2008) and
a precursor for tropospheric $O_3$ (Steinfeld, 1998). Being the major sink of OH, CO also controls the atmosphere's oxidizing
capacity (Levy, 1971;Novelli et al., 1998). As the only primary alkaline gas in the atmosphere, $NH_3$ is closely associated with
the acidity of precipitations for one thing, for another it can react with sulfuric acid and nitric acid forming ammonium sulfate
and ammonium nitrate which account for a large proportion of fine particulate matter (Sun et al., 2012;Sun et al., 2013).
Assessing their model performances is thus important to help us better understand their environmental consequences and also
help explain the model performances for their related secondary air pollutants, such as $O_3$ and fine particulate matter.
In previous phase of MICS-Asia, no specific evaluation and inter-comparison work has been conducted for these gases,
especially for CO and $NH_3$. In MICS-Asia II, model performance of $NO_2$ was evaluated as a relevant species to $O_3$ (Han et al.,
2008b), however such evaluations were limited to the observation sites from EANET (Acid Deposition Monitoring Network
in East Asia). Model evaluations and inter-comparisons in industrialized regions of China has not been performed due to the
limited number of monitoring sites in China from EANET, which hindered our understanding of the model performance in
industrialized regions. More densely observations over highly industrialized regions of China, namely the North China (NCP)
Plain and Pearl River Delta (PRD) regions, were first included in MICS-Asia III, allowing the model evaluations over highly
industrialized regions. Meanwhile, the emission inventories of these three gases still subject to the large uncertainties
(Kurokawa et al., 2013;Li et al., 2017b), which is a major source of uncertainties in air quality modeling and forecast.
Evaluating these gases' emission inventories from a model perspective is also a useful way to identify the uncertainties in
emission inventories (Han et al., 2008a;Noije et al., 2006;Pinder et al., 2006;Stein et al., 2014;Uno et al., 2007).
In all, this paper is aimed at evaluating the $NO_2$, CO and $NH_3$ simulations using the multi-model data from MICS-Asia
III, three questions are trying to be addressed: (1) what is the performance of current CTMs in simulating the $NO_2$, CO and
$NH_3$ concentrations over highly industrialized regions of China, (2) what are the potential factors responsible for the model
deviations from observations and differences among models, and (3) how large are the impacts of model uncertainties on the
simulations of these gases.
**2 Inter-comparison frameworks**
**2.1 Description on the participating models and input datasets**
Six different chemical transport models have participated in MICS-Asia III with their major configurations summarized
in Table 1. These models included NAQPMS (Wang et al., 2001), three versions of CMAQ (Byun and Schere, 2006), WRF-
Chem (Grell et al., 2005), NU-WRF (Peters-Lidard et al., 2015), NHM-Chem (Kajino et al., 2012) and GEOS-Chem
(http://acmg.seas.harvard.edu/geos/). All models employed a same modeling domain (Fig. 1) with a horizontal resolution of
45km except M13 (0.5° of latitude×0.667° of longitude) and M14 (64km×64km). Detailed information on each component of
these CTMs can be obtained from the companion paper Chen et al., 2019 and Tan et al., 2019.
Standard model input datasets of raw meteorological fields, emission inventory and boundary conditions were provided
by MICS-Asia III for all participants. Raw meteorological fields were generated from a whole year simulations of 2010 using
Weather Research and Forecasting Model (WRF) version 3.4.1 (Skamarock, 2008) with horizontal resolution of 45km and
vertically 40 layers from surface to the model top (10hPa). Initial and lateral boundary conditions for meteorological simulation
were generated every six hours by using the 1°×1° NCEP FNL (Final) Operational Global Analysis data (ds083.2). Real-time,
global, sea surface temperature (RTG_SST_HR) analysis were used to generate and update lower boundary conditions for sea
areas. Four-dimensional data assimilation nudging (Gridded FDDA & SFDDA) was performed during the simulation to
increase the accuracy of WRF after the objective analysis with NCEP FNL (Final) Operational Global Analysis data (ds083.2),
NCEP ADP Global Surface Observation Weather Data (ds461.0) and NCEP ADP Global Upper Air and Surface Weather Data
(ds337.0). Detailed configurations of the standard meteorological model are available in supplementary Table S1. The
simulated wind speed, relative humidity and air temperature were evaluated against the observations over the NCP and PRD
regions with detailed results shown in supplementary Sect. S1. In general, the standard meteorological simulations well
captured the main features of meteorological conditions in the NCP and PRD regions with high correlation coefficient, small
biases and low errors for all meteorological parameters (supplementary Fig.S1-S3 and Table S2).

Standard emission inventories provided by the MICS-Asia III were used by all participants. The anthropogenic emissions

were provided by a newly developed anthropogenic emission inventory for Asia (MIX) which integrated five national or
regional inventories, including Regional Emission inventory in Asia (REAS) inventory for Asia developed at the Japan
National Institute for Environment Studies, the Multi-resolution Emission Inventory for China (MEIC) developed at Tsinghua
University, the high-resolution ammonia emission inventory in China developed at Peking University, the Indian emission
inventory developed at Argonne National Laboratory in the United States, and the Clean Air Policy Support System (CAPSS)
Korean emission inventory developed at Konkuk University (Li et al., 2017b). Hourly biogenic emissions for the entire year
of 2010 in MICS-Asia III were provided by the Model of Emissions of Gases and Aerosols from Nature version 2.04 (Guenther
et al., 2006). The Global Fire Emissions Database 3 (Randerson et al., 2013) was used for biomass burning emissions. Volcanic
$SO_2$ emissions were provided by the Asia Center for Air Pollution Research (ACAP) with a daily temporal resolution. Air and
ship emissions with an annual resolution were provided by the HTAPv2 emission inventory for 2010 (Janssens-Maenhout et
al., 2015). NMVOC emissions were spectated into the model-ready inputs for three chemical mechanisms: CBMZ, CB05 and
SAPRC-99 and the  weekly and diurnal profiles for emissions were also provided.

MICS-Asia III has provided two sets of top and lateral boundary conditions for year 2010, which were derived from the

3-hourly global CTM outputs of CHASER (Sudo et al., 2002a; Sudo et at., 2002b) and GEOS-Chem
(http://acmg.seas.harvard.edu/geos/), run by Nagoya University (Japan) and the University of Tennessee (USA) respectively.
GEOS-Chem was run with 2.5º×2º resolution and 47 vertical layers while CHASER model was run with 2.8º×2.8º and 32
vertical layers.

All participants were required to use the standard model input data to drive their model run so that impacts of model input

data on simulations could be minimized. However, models are quite different from each other, and it is difficult to keep all the
inputs the same. The majority of models have applied the standard meteorology fields, while the GEOS-Chem and RAMS-
CMAQ utilized their own meteorology models. The GEOS-Chem was driven by the GEOS-5 assimilated meteorological fields
from the Goddard Earth Observing System of the NASA Global Modeling Assimilation Office, and the RAMS-CMAQ was
driven by meteorological fields provided by Regional Atmospheric Modeling System (RAMS) (Pielke et al., 1992). WRF-
Chem utilized the same meteorology model (WRF) as the standard meteorological simulation, but two of them considered the
two-way coupling effects of pollutants and meteorological fields. The meteorological configurations of these WRF-Chem
models were compared to the configurations of the standard meteorological model (supplementary table S1), which shows
slight differences from the standard meteorological model. The CTM part of NHM-Chem is coupled with the JMA's non-
hydrostatic meteorological model (NHM) (Saito et al., 2006), but an interface to convert a meteorological model output of
WRF to a CTM input was implemented (Kajino et al., 2018). Thus, the standard meteorology field was used in the NHM-
Chem simulation, too.

## 2.2 Data and statistical methods

All modeling groups have performed a base year simulations of 2010 and were required to submit their modeling results
according to the data protocol designed in MICS-Asia III. Gridded monthly concentrations of $NO_2$, CO, $NH_3$ and ammonium
($NH_4^+$) in the surface layer were used in this study. Note that modeling results from M3 and $NH_3$ simulations from M8 were
excluded due to their incredible results, thus only thirteen modeling results were used in this study.
Hourly observed concentrations of $NO_2$ and CO were collected over the NCP (19 stations) and PRD (13 stations) regions,
obtained from the air quality network over North China (Tang et al., 2012) and the Pearl River Delta regional air quality
monitoring network (PRD RAQMN), respectively. The air quality monitoring network over North China was set up by the
Chinese Ecosystem Research Network (CERN), the Institute of Atmospheric Physics (IAP) and the Chinese Academy of
Sciences (CAS) since 2009 within an area of $500 \times 500$ km$^2$ in northern China. All monitoring stations were selected and set
up according to the US EPA method designations (Ji et al., 2012). The PRD RAQMN network was jointly established by the
government of the Guangdong Province and the Hong Kong Special Administrative Region, consisting of 16 automatic air
quality monitoring stations across the PRD region (Zhong et al., 2013). Thirteen of these stations are operated by the
Environmental Monitoring Centers in the Guangdong Province which were used in this study, while the other three are located
in Hong Kong (not included in this study) and are managed by the Hong Kong Environmental Protection Department. Monthly
averaged observations were calculated for the comparisons with the simulated monthly surface $NO_2$ and CO concentrations. It
should be noted that these networks measured the $NO_2$ concentrations using a thermal conversion method, which would
overestimate the $NO_2$ concentrations due to the positive interference of other oxidized nitrogen compounds (Xu et al., 2013).
$NH_3$ observations for long-term period are indeed challenging and limited due to its strong spatial and temporal variability,
quick conversion from one phase to another and also its stickiness to the observational instruments (von Bobrutzki et al., 2010).
Measurements of surface $NH_3$ concentrations in year 2010 were not available in this study, however, one-year surface
measurement of monthly $NH_3$ concentrations over China from September of 2015 to August of 2016 were used as a reference
dataset in this study, which were obtained from the Ammonia Monitoring Network in China (AMoN-China) (Pan et al., 2018)
The AMoN-China was established based on the CERN and the Regional Atmospheric Deposition Observation Network in
North China Plain (Pan et al., 2012), which consists of 53 sites over the whole China and measured the monthly ambient $NH_3$
concentrations using the passive diffusive technique. Eleven stations located in the NCP region were used in this study.
Distributions of the observation sites of $NO_2$, CO and $NH_3$ over the NCP and PRD regions as well as their total emissions in
year 2010 provided by MICS-Asia III are shown in Fig. 1. Besides the surface observations, the satellite retrievals of $NH_3$ total
columns from IASI (Infrared Atmospheric Sounding Interferometer) were also used in this study to quantitatively evaluate the
modeled monthly variations of $NH_3$ concentrations. The ANNI-NH3-v2.1R-I retrieval product (Van Damme et al., 2017;Van
Damme et al., 2018) was used in this study which is the reanalysis version of $NH_3$ retrievals from IASI instruments and
provides the daily morning (~9:30 am local time) $NH_3$ total columns from year 2008 to 2016. More detailed information and
the process of satellite data are available in supplementary sect. S2.
Mean bias error (MBE), normalized mean bias (NMB), root mean square error (RMSE) and correlation coefficient (R)
were calculated for the assessment of model performances. Standard deviation of the ensemble models was used to measure
the ensemble spread and the impacts of model uncertainty. Coefficient of variation (hereinafter, CV), defined as the standard
deviation divided by the average with larger value denoting lower consistency among models, was also used to measure the
impacts of model uncertainty in a relative sense. However, by this definition, there is a tendency that lower concentrations are
more likely associated with higher value of CV, thus we did not calculate the values of CV over model grids whose simulated
concentrations were lower than 0.1 ppbv for $NO_2$ and $NH_3$, and 0.1 ppmv for CO, respectively. March–May, Jun–August,
September–November and December–February were used to define the four seasons that are spring, summer, autumn and
winter, respectively.
**3 Results**
**3.1 Evaluating the ensemble models with observations**
To facilitate comparisons, the modeling results were interpolated to the observation sites by taking the values from the
grid cell where the monitoring stations located. Model evaluation metrics defined in Sect. 2.2 were then calculated to evaluate
the modeling results against the observations.
**3.1.1 $NO_2$**
Figure 2 displays the comparisons between the observed and simulated annual mean $NO_2$ concentrations over the NCP
(2a) and PRD(2b) regions with calculated model evaluation metrics summarized in Table 2. M13 is not included in the
evaluation of $NO_2$ since it did not submitted the $NO_2$ concentrations. In general, the majority of models underpredicted $NO_2$
levels in both the NCP and PRD regions. Calculated MBE (NMB) ranges from -6.54 ppbv (-28.4%) to -2.45 (-10.6%) ppbv
over the NCP region and from -9.84 ppbv (-44.0%) to -1.84 ppbv (-8.2%) over the PRD regions among these negatively-biased
models. These underpredicted $NO_2$ concentrations are consistent with the overpredicted $O_3$ concentrations by these models
found in the companion paper by Li et al., 2019. $O_3$ productions can either increase with $NO_x$ under $NO_x$ limited conditions or
decrease under the $NO_x$ saturated (also called volatile organic compounds (VOCs) limited) conditions (Sillman, 1999). Both
the NCP and PRD regions are industrialized regions in China with high $NO_x$ emissions (Fig. 1). Observations also showed that
the NCP and PRD regions are falling into or changing into the $NO_x$ saturated regimes (Shao et al., 2009;Jin and Holloway,
2015). Therefore, the underestimated $NO_2$ concentrations may contribute to the overpredicted $O_3$ concentrations in these two
regions. More details about the $O_3$ predictions can be found in the companion paper by Li et al., 2019. In addition, as we
mentioned in Sect.2.2, the negative biases in the simulated $NO_2$ concentrations can be also partly attributed to the positive
biases in the $NO_2$ observations. M5, M8, M9 and M11 in the NCP region and M5, M8 and M11 in the PRD region were
exceptions that overpredicted $NO_2$ concentrations. M11 showed good performances in predicting $NO_2$ levels in the NCP region
with smallest RMSE, while M9 significantly overestimated $NO_2$ with largest MBE and RMSE values. $NO_2$ predictions by M8
were close to the observations over the PRD region with smallest RMSE value. Meanwhile, we also found that models
exhibited better $NO_2$ modeling skills in the NCP region than that in the PRD region with smaller bias and RMSE values.

According to the spatial correlation coefficients (Table 2), all models well reproduced the main features of the spatial

variability of $NO_2$ concentrations in the NCP region with correlation coefficients ranging from 0.57 to 0.70. However, models
failed in capturing the spatial variability of $NO_2$ concentrations in the PRD region with correlation coefficients only ranged
from 0.00 to 0.38. Such low correlation might be attributed to the coarser model resolution (45km) that some local impacts on
the $NO_2$ concentrations might not be well resolved in the model, and/or related to the uncertainties in emission inventories
which were not well resolved in the PRD region. To investigate it, we have conducted an additional one-year simulation with
finer horizontal resolutions (15km and 5km, supplementary Fig.S4) in the PRD region using the NAQPMS model. Detailed
experimental settings are presented in supplementary Sect.S3. The experiment results indicate that when using the same
emission inventory as the coarse-resolution simulation, the high-resolution simulation still show poor model performances in
capturing the spatial variability of $NO_2$ concentrations in the PRD region, with calculated correlation coefficient only of 0.03
and 0.02 for 15km and 5km resolutions, respectively ( supplementary Sect. S3, Fig. S5-6 and Table S3). Thus, the poor model
performance in the PRD region could be more related to the coarse resolution and/or inappropriate spatial allocation of the
emission inventories. These results also suggested that only increasing the resolutions of model may not help improve the
model performance.

Figure 3 presents the monthly timeseries of the observed and simulated regional mean $NO_2$ concentrations over the NCP

(3a) and PRD (3b) regions from January to December in 2010. The models well captured the monthly variations of $NO_2$
concentrations both in the NCP and PRD regions. According to Table 2, the correlation coefficient ranges from 0.28 to 0.96
in the NCP region and from 0.52 to 0.95 in the PRD region. M8 showed the largest overestimation among all models in summer
that MBE (NMB) can reach 12.1 ppbv (75.8%) in the NCP region, which may help explain the low correlation of this model.
M9 exhibited a significant overestimation in winter in the NCP region with MBE (NMB) up to 22.0 ppbv (79.3%) while much
less overestimation or even underestimation (summer) in other seasons. This discrepancy may be explained by that M9 was
an online coupled model which considers two-way coupling effects between the meteorology and chemistry. During the period
with heavy haze, the radiation can be largely reduced by aerosol dimming effects, leading to weakened photochemistry,
lowered boundary layer height and thus the increase of $NO_2$ concentrations. Severe haze was reported to occur in North China
in January 2010, with maximum hourly $PM_{2.5}$ concentration even reached as high as ~500 $\mu g/m^3$ in urban Beijing (Gao et al.,
2018). Such high aerosol loadings in atmosphere could trigger interactions between chemistry and meteorology. Interestingly,
M9 did not overestimate $NO_2$ during winter in the PRD region. This might be related to the lower aerosol concentrations and
weaker chemistry-and-meteorology coupling effects in the PRD region.
**3.1.2 CO**

Similar analyses were performed for modeling results of CO. All models significantly underestimated the annual mean

CO concentrations both in the NCP and PRD regions (Figs. 2c-d and Table 2). Calculated MBE (NMB) ranges from -1.69
ppmv (-76.2%) to -1.16 ppmv (-52.0%) in the NCP region and from -0.67 ppmv (-69.6%) to -0.50 ppmv (-52.3%) in the PRD
region (Table 2). Such large negative biases in all models were not likely to be explained by the model uncertainties, suggesting
the negative biases in the CO emissions over China. This is consistent with the inversion results of Tang et al., 2013 which
indicates a significant underestimation of CO emissions over the Beijing and surrounding areas in the summer of 2010. Over
the latest decades, global models also reported CO underestimations in north hemisphere (Naik et al., 2013;Stein et al., 2014)
and a number of global model inversion studies have been conducted to derive the optimized CO emissions. Most of these
studies have reported a significant underestimation of CO emissions in their *a priori* estimates (Bergamaschi et al.,
2000;Miyazaki et al., 2012;Petron et al., 2002;Petron et al., 2004). Our findings agree with these studies and indicate that more
accurate CO emissions are needed in future studies. Model performances in simulating spatial variability of CO concentrations
were still poor in the PRD region according to Table 2 with most models showing negative correlation coefficients.

Timeseries of the observed and simulated regional mean CO concentrations in the NCP and PRD regions are presented

in Fig.3c-d. It shows that the models well reproduced the monthly variations of CO concentrations in both the NCP and PRD
regions with high temporal correlation coefficient except M5 (Table 2). All models, however, underestimated CO
concentrations throughout the year and showed largest underestimations in winter with MBE (NMB) by ensemble mean up to
-2.1 ppmv (-64.9%) in the NCP region and -0.75 ppmv (-60.6%) in the PRD region.
**3.1.3 NH$_3$**

Figure 2e shows the comparisons of the observed and simulated annual mean NH$_3$ concentrations in the NCP region.

Since we used the NH$_3$ observations from September 2015 to August 2016, negative biases are expected according to the
increasing trend of atmospheric ammonia during period 2003–2016 detected by recently retrievals from the Atmospheric
Infrared Sounder (AIRS) aboard NASA's Aqua satellite (Warner et al., 2016;Warner et al., 2017). Due to the interannual
uncertainty, we mainly focused on the disparities among different models rather than the deviation from observations.

Large differences can be seen in simulated NH$_3$ concentrations from different models. M14 simulated very low

concentrations and exhibited the largest negative biases with MBE (NMB) of -12.2 ppbv (-66.3%), which may be related to
the higher conversion rate of NH$_3$ to NH$_4^+$ in M14 (discussed in later part of this section). On the contrary, M9 provided much
higher NH$_3$ concentrations than other models with MBE (NMB) up to 21.8 ppbv (118.7%). For the CMAQ models, M1 and
M2 exhibited higher NH$_3$ concentrations and larger spatial variability compared to other CMAQ models. Such discrepancy
may be explained by that M1 and M2 are two model runs using CMAQ v5.0.2. The bi-directional exchange of NH$_3$ has been
integrated into CMAQ from version 5.0. This module can simulate the emitted and deposited processes of NH$_3$ between
atmosphere and the surfaces, allowing the additional $NH_3$ emissions to the atmosphere (US EPA Office of Research and
Development).
As can be seen in Table 2, the observed spatial variations of $NH_3$ over the NCP region can be well reproduced by all
models (R = 0.57-0.71), indicating that the spatial variations of current $NH_3$ emissions over the NCP region are well represented
in emission inventories. However, all models failed to capture the observed monthly variations of $NH_3$ concentrations with
most models mismatching the observed $NH_3$ peak (July) and showing negative correlation coefficients. M10 and M13 are
exceptions showing good temporal correlations of 0.64 and 0.65, respectively (Fig. 3e and Table 2). This is quite different
from the model behavior in simulating the monthly variations of $NO_2$ and CO concentrations. As seen in Fig. 3e, the
observation showed the peak concentrations of $NH_3$ in summer months and lower concentrations in autumn and winter, which
is consistent with the previous $NH_3$ observations in the NCP region (Shen et al., 2011;Xu et al., 2016;Meng et al., 2011).
Newly derived satellite-measured $NH_3$ at 918 hPa averaged between September 2002 and August 2015 also demonstrated
higher concentrations in spring and summer and lower concentrations in autumn and winter (Warner et al., 2016). However,
all models predicted a peak concentration in November except M10 in August in and M13 in June. We also used the satellite
retrievals of $NH_3$ total columns from IASI to further evaluate the modeled monthly variations of $NH_3$ concentrations, since
evaluating the model results using observations from different years may be inappropriate due to the emission change of $NH_3$.
Comparisons of the surface $NH_3$ observations from AMoN-China and $NH_3$ total columns form IASI (supplementary Fig.S7)
suggest that the IASI measurement can well represent the monthly variations of surface $NH_3$ concentrations, which can be
used to qualitatively evaluate the modeled monthly variations of surface $NH_3$ concentrations. The monthly time series of the
regional mean $NH_3$ total columns over the NCP region from January, 2008 to December, 2016 are shown in supplementary
Fig. S8, which shows similar monthly variations to the surface $NH_3$ observations with highest value in July and confirms the
poor model performances in reproducing the monthly variations of $NH_3$ concentrations. The IASI measurement also indicates
that the interannual variability of monthly variations of $NH_3$ concentrations over the NCP region was small from year 2008 to
2016, which suggest that using observations from different years could still provide valuable clues for verifying the modeled
monthly variations.
The simulated monthly variations of $NH_3$ concentrations were closely related to the monthly variations of the $NH_3$
emissions. Most models predicted three peak values of $NH_3$ concentrations in June, August and November but exhibited a
significant decrease in July, which was in good agreement with the peaks and drops of the $NH_3$ emission rates in these months
(Fig.4). The strong relationship between the simulated $NH_3$ concentrations and the emission rates suggests that the poor model
performance in reproducing the monthly variations of $NH_3$ concentrations is probably related to the uncertainties in the monthly
variations of $NH_3$ emissions. This is consistent with the recent bottom-up and top-down estimates of agriculture ammonia
emissions in China by (Zhang et al., 2018), which shows more distinct seasonality of Chinese $NH_3$ emissions.
It is worth noting that there are also important uncertainties in the models beyond emission uncertainty. In order to
investigate this issue, we have analyzed the impact of gas-aerosol partitioning of $NH_3$ on the simulations of $NH_3$ concentrations.
Figure 5 shows the timeseries of the simulated total ammonium ($NH_x = NH_3 + NH_4^+$) in the atmosphere along with the ratio

of gaseous $NH_3$ to the total ammonium. M10 is excluded in Fig.5 since the GOCART model does not predict $NH_4^+$ concentrations. As a result, the emitted $NH_3$ would be only presented as the gas phase in M10, leading to higher $NH_3$ predictions. This may also help explain the different monthly variations of $NH_3$ concentrations seen in M10. Without the considerations of $NH_4^+$, the monthly variations of $NH_3$ concentrations in M10 were more consistent with the monthly variations of $NH_3$ emissions, which highlighted the importance of gas-aerosol partitioning of $NH_3$ on the predictions of monthly variations of $NH_3$ concentrations. As seen in fig.5, there are large discrepancy in the simulated gas-aerosol partitioning of $NH_3$ from different models. M7 and M9 showed higher $NH_3/NH_x$ ratio than other models, which means that these two models tended to retain the $NH_3$ in the gas phase and thus predicted higher $NH_3$ concentrations than other models. For example, M7 predicted comparable magnitude of total ammonium with most models, while gas $NH_3$ concentration in M7 accounted for more than 60% of total ammonium in summer and even 90% in winter. The lower conversion rate of $NH_3$ to $NH_4^+$ in M9 may be related to the gas phase chemistry used in the model. M9 used the RADM2 mechanism which gives lower reaction rates of oxidation of $SO_2$ and $NO_2$ by the OH radical as compiled by Tan et al., 2019, leading to lower productions of acid and thus lower conversion rate of $NH_3$ to $NH_4^+$. In case of M7, the hydrolysis of $N_2O_5$ was not considered in M7, which leads to a lower tendency in the prediction of $NO_3^-$ (Chen et al., 2019) and partly explains the higher $NH_3$ predictions of M7. On the contrary, M14 showed a much lower $NH_3/NH_x$ ratio than most models, which would be related to its higher production rates of sulfate than other models as seen in Chen et al., 2019. In terms of monthly variations, most models predicted lower $NH_3/NH_x$ ratio in summer than that in other seasons, suggesting the higher conversion rates of $NH_3$ from gas phase to aerosol phase in summer. This would be related to the higher yield of ammonium sulfate due to the enhanced photochemical oxidation activity in summer. However, different from the modeling results, the $NH_3$ and $NH_4^+$ observations over the NCP region indicated a lower $NH_3/NH_x$ ratio with higher ammonium concentrations in autumn and winter (Shen et al., 2011;Xu et al., 2016). Although observed $NH_4^+$ was largest in summer at a rural site in Beijing, observed $NH_3/NH_x$ ratio was still highest in summer according to observations from Meng et al., 2011. These results indicate that there would be large uncertainties in the modeling of seasonal variations of the gas-aerosol partitioning of $NH_3$ over the NCP region. The formation of $NH_4^+$ mainly depends on the acid gas concentrations, temperature, water availability (Khoder, 2002) and the flux rates of $NH_3$ (Nemitz et al., 2001). Compared with spring and summer, the lower temperature and higher $SO_2$ and $NO_x$ emissions should favor the gas-to-particle phase conversion of $NH_3$ and lead to higher $NH_4^+$ concentrations. This contrast indicates that some reaction pathways of acid productions ($H_2SO_4$ or $HNO_3$) may be missing in current models, such as aqueous-phase and heterogeneous chemistry (Cheng et al., 2016;Wang et al., 2016;Zheng et al., 2015). Such uncertainty may be another important factor contributing to the poor model performances in reproducing the monthly variations of $NH_3$ concentrations over the NCP region.

## 3.2 Quantifying the impacts of model uncertainty

In this section, we further investigate the discrepancies among the different models to quantify the impacts of model uncertainty on the simulations of these gases. As we mentioned in Sect. 2, most of these models employed common

meteorology fields and emission inventories over China under the same modeling domain and horizontal resolutions, which composed an appropriate set for the investigations of model uncertainties.

Figures 6–8 present the simulated annual mean concentrations of $NO_2$, CO and $NH_3$ from different models. The spatial distributions of the simulated $NO_2$, CO and $NH_3$ concentrations from different models agreed well with each other, similar to the spatial distributions of their emissions (Fig. 1). High $NO_2$ concentrations were mainly located in the north and central-east China, and several hot-spots of $NO_2$ were also detected in the northeast China and the PRD region. M5, M8, M9, and M11 predicted higher $NO_2$ concentrations than other models especially for M8 which also predicted very high $NO_2$ levels over southeast China. Similar to $NO_2$, high CO concentrations were generally located over the north and central-east China as well as the east of Sichuan basin. M8, M9 and M11 predicted higher CO concentrations than other models as well. In terms of $NH_3$, although most models shared similar spatial patterns of $NH_3$ simulations, the simulated $NH_3$ concentrations varied largely from different models. High $NH_3$ concentrations were mainly located over the north China and India peninsula, which was in accordance with the distribution of agricultural activity intensity over East Asia. Among these models, M9 and M10 produced much higher $NH_3$ concentrations over East Asia while M4, M5, M6, M13 and M14 produced much lower concentrations.

The impacts of model uncertainty on the simulations of $NH_3$ (9a), CO (9b) and $NO_2$ (9c) were then quantified in Fig.9, denoted by the spatial distributions of the standard deviation (ensemble spread) and the corresponding distributions of CV on the annual and seasonal basis. Note that M13 and M14 are excluded in the calculation of ensemble spread and CV to reduce the influences of the meteorological input data and horizontal resolutions. It seems that the impacts of model uncertainty increase with the reactivity of gases. $NH_3$ simulations were affected most by the model uncertainty, while CO suffered least from the uncertainty in models.

The ensemble spread of $NH_3$ simulations exhibited a strong spatial variability with higher values mainly located in the NCP region. Standard deviation of the annual mean $NH_3$ concentrations can be over 20 ppbv in Henan province and 15 ppbv in the south of Hebei province, which is about 60–80% and 40–60% of the ensemble mean respectively according to the CV distribution. As we mentioned in Sect. 3.1.3, these large modeling differences can be partly explained by the differences in the bi-directional exchange and gas-aerosol partitioning of $NH_3$ in different models. A strong seasonal pattern was also found in the differences of $NH_3$ simulations over the NCP region. The ensemble spread was smallest in spring while largest in autumn, up to 25 ppbv in most areas of the NCP region. However, in the relative sense, the modeling differences were larger in summer and winter while less in spring and autumn. The southeast China shared a similar magnitude of the ensemble spread (2–5 ppbv) and showed weaker seasonal variability. However, the modeling differences in the relative sense were larger than that in the NCP region with CV over 1.0 in all seasons except that in Summer. This can be due to that the simulated concentrations may be more influenced by the model processes over the areas with low emissions, while more constrained by the emissions over high emission rate areas.

CO was least affected by the model uncertainty among the three gases which is consistent with its weaker chemical activity and longer lifetime in the atmosphere. The ensemble spread of annual mean CO concentration was about 0.05–0.2 ppmv in the east China, only about 20%–30% of the ensemble mean. Meanwhile, CO modeling differences was more

uniformly distributed in east China with CV less than 0.3 over most areas of east China. However, large modeling differences
were visible over Myanmar during spring when there were high CO emissions from biomass burning. Model differences turned
to be larger during winter in the NCP region with ensemble spread and CV about 0.3–0.5 ppmv and 0.3–0.4, respectively.
$NO_2$ was mediumly affected by the model uncertainty among the three gases. Ensemble spread of annual mean $NO_2$
concentration was 5–7.5 ppbv in the NCP region and 2.5–5 ppbv in the southeast China, which accounted for about 20%–30%
of the ensemble mean in the former but more than 70% in the latter. The ensemble spread was largest in winter which was
over 10 ppbv in the NCP region (30%–40%) and 5–7.5 ppbv in southeast China (over 70%). Similar to $NH_3$, southeast China
exhibited more modeling differences than the NCP region in relative sense with CV higher than 0.7 in most areas of southeast
China.

## 409 4 Summary

In this study, thirteen modeling results of surface $NO_2$, CO and $NH_3$ concentrations from MICS-Asia III were compared
with each other and evaluated against the observations over the NCP and PRD regions. Three questions are trying to be
addressed which are related to the performance of current CTMs in simulating the $NO_2$, CO and $NH_3$ concentrations over the
highly industrialized regions of China, potential factors responsible for the model deviations from observations and differences
among models, and the impacts of model uncertainty on the simulations of these gases.
Most models showed underestimations of $NO_2$ concentrations in the NCP and PRD regions, which could be an important
potential factor contributing to the overpredicted $O_3$ concentrations in these regions. According to Xu et al., 2013, such
underestimations would also be related to the positive biases in the $NO_2$ observations. The models showed better $NO_2$ model
performance in the NCP region than that in the PRD region with smaller biases and RMSE. Most models well reproduced the
observed temporal and spatial patterns of $NO_2$ concentrations in the NCP region, while relatively poor model performance was
found in the PRD region in terms of the spatial variations of $NO_2$ concentrations. A sensitivity test with finer horizontal
resolutions has been conducted to investigate the potential reasons for the poor model performance in the PRD region. The
results shows that only increasing the model resolution cannot improve the model performance in the PRD region, which
suggest that the poor model performance in the PRD region would be more related to the coarse resolution and/or inappropriate
spatial allocations of the emission inventories in the PRD regions. All models significantly underestimated the CO
concentrations in the NCP and PRD regions throughout the year. Such large underestimations of all models are not likely to
be fully explained by the model uncertainty, which suggests that CO emissions may be underestimated in current emission
inventories. More accurate estimate of CO emissions is thus needed for year 2010. Underestimations of CO emissions may be
alleviated in recent years due to the decreasing trends of the Chinese CO emissions in recent years(Jiang et al., 2017;Zhong et
al., 2017;Sun et al., 2018;Muller et al., 2018;Zheng et al., 2018;Zheng et al., 2019). The inversion results of Zheng et al., 2018
also agree well with the MEIC inventory for CO emissions in China from 2013 to 2015. However uncertainties still exist in
the CO emissions for recent years, according to previous studies, the estimated CO emissions in China ranges from 134–202
Tg/yr in year 2013 (Jiang et al., 2017;Zhong et al., 2017;Sun et al., 2018;Muller et al., 2018;Zheng et al., 2018;Zheng et al.,
2019). Zhao et al., 2017 also suggested a -29%–40% uncertainty of CO emissions from the industrial sector in year 2012. For
$NH_3$ simulations, in contrast to the good skills in the monthly variations of $NO_2$ and CO concentrations, all models failed to
reproduce the observed monthly variations of $NH_3$ concentrations in the NCP region, as shown by both the surface and satellite
measurements. Most models mismatched the observed peak and showed negative correlation coefficient with observations,
which may be closely related to the uncertainty in the monthly variations of $NH_3$ emissions and also the uncertainty in the gas-
aerosol partitioning of $NH_3$.
Several potential factors were found to be responsible for the model deviation and differences, including the emission
inventories, chemistry-and-meteorology coupling effects, bi-directional exchange of $NH_3$ and the $NH_3$ gas-aerosol partitioning,
which would be important aspects with respect to the model improvements in future. Previous studies also suggest that the
nitrous acid (HONO) chemistry plays an important role in the atmospheric nitrogen chemistry, which influences the
simulations of $NO_2$ and $NH_3$ (Fu et al., 2019;Zhang et al., 2017;Zhang et al., 2016). Heterogeneous conversion from $NO_2$ to
HONO ($2NO_{2(g)} + H_2O_{(l)} \rightarrow HONO_{(l)} + HNO_{3(l)}$) is one of the dominant sources of HONO in the atmosphere, which has been
considered in most models of MICS-Asia III, including CMAQ since version 4.7, NAQPMS, NHM-Chem and GEOS-Chem.
However, some other important sources of HONO may still be underestimated by models in MICS-Asia III. For example, Fu
et al., 2019 suggested that the high relative humidity and strong light could enhance the heterogeneous reaction of $NO_2$ , and
the photolysis of total nitrate were also important sources of HONO. These sources has not been included in the models of
MICS-Asia III, which would lead to the deviations from observations. The inter-comparisons of the ensemble models
quantified the impacts of model uncertainty on the simulations of these gases, which shows that the impacts of model
uncertainty increases with the reactivity of these gases. Models contained more uncertainties in the prediction of $NH_3$ than the
other two gases. Based on these findings, some recommendations are made for future studies:
1) More accurate estimation of CO and $NH_3$ emissions are needed in future studies. Both bottom-up and top-down method
(inversion technique) can help address this problem. The inversion of $NH_3$ emissions would be more complicated than the
inversion of CO emissions due to the larger uncertainties in modeling the atmospheric processes of $NH_3$. Nevertheless, it could
still provide valuable clues for verifying the bottom-up emission inventories (Zhang et al., 2009) if the models were well
validated. In addition, by using the ground or satellite measurements, the top-down methods could also give valuable
information on the spatial and temporal patterns of $NH_3$ emissions, for example the inversions studies by Paulot et al., 2014
and Zhang et al., 2018. However, more attention should be paid to the validations of model before the inversion estimation of
$NH_3$ emissions. How to represent the model uncertainties in the current framework of emission inversion is also an important
aspect in future studies. Things could be better for CO considering its small and weakly spatial-dependent model uncertainties.
2) For some highly active and/or short-lived primary pollutants, like $NH_3$, model uncertainty can also take a great part in
the forecast uncertainty. Emission uncertainty alone may not be sufficient to explain the forecast uncertainty and may cause
underdispersive, and overconfident forecasts. Future studies are needed in how to better represent the model uncertainties in
the model predictions to obtain a better forecast skill. Such model uncertainties also emphasize the need to validate the
individual model before using its results to make important policy recommendation.

3) Gas-aerosol partition of $NH_3$ is shown to be an important source of uncertainties in $NH_3$ simulation. The formation of

$NH_4^+$ particles is mainly limited by the availability of $H_2SO_4$ and $HNO_3$ under ammonia-rich conditions, which involves
complex chemical reactions, including gas-phase, aqueous-phase and heterogeneous chemistry (Cheng et al., 2016;Wang et
al., 2016;Zheng et al., 2015). These processes are needed to be verified and incorporated into models to better represent the
chemistry in the atmosphere.

4) The gas chemistry mechanisms used in this study are SAPRC 99, CB05, CBMZ, RACM and RADM2, and some of

them have an updated version such as CB06 and SPARC 07. Our conclusions may not be applicable to these newer versions
of mechanisms and thus more comparisons studies can be performed to understand the differences in these new mechanisms.

**Competing interests**

The authors declare that they have no conflict of interest.

**Author contribution**

X.T., J.Z., Z.F.W and G.C. conducted the design of this study. J.F., X.W., S.I., K.Y., T.N., H.L., C.K., C.L., L.C., M.Z., Z.T.,
J.L., M.K., H.L., B.G. contributed to the modelling data. Z.W. performed the simulations of standard meteorological field.
M.L. and Q.W. provided the emission data. K.S. provided the CHASER output for boundary conditions. Y.W., Y.P., G.T.
provided the observation data. L.K. and X.T. performed the analysis and prepared the manuscript with contributions from all-
authors.

**Acknowledgements**

This study was supported by the National Natural Science Foundation (Grant Nos. 91644216 & 41620104008), the National
Key R&D Program (Grant Nos. 2018YFC0213503) and Guangdong Provincial Science and Technology Development Special
Fund (No.2017B020216007). Yuepeng Pan acknowledges the National Key Research and Development Program of China
(Grants 2017YFC0210100, 2016YFC0201802) and the National Natural Science Foundation of China (Grant 41405144) for
financial support. We are indebted to the staff who collected the samples at the AMoN-China sites during the study period.

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

**Tables**
**Table 1: Basic configurations of participating models in MICS-Asia III**

| No | Horizontal resolution | Vertical resolution | First layer height | Horizontal advection | Vertical advection | Horizontal Diffusion | Vertical Diffusion | Gas phase chemistry | Aerosol processes | Dry depositiono f gases | Wet deposition of gases | Meteorology | Boundary condition | Online (Yes or No) |
|---|---|---|---|---|---|---|---|---|---|---|---|---|---|---|
| M1 | 45km | $40\sigma_p$ level | 57 m | Yamo (Yamartino, 1993) | ppm (Collella and Woodward, 1984) | multiscale | ACM2 (Pleim, 2007) | SAPRC99 (Carter, 2000) | Aero6 (Binkowski and Roselle, 2003) | Wesely (1989) | Henry's law | Standard[a] | GEOS-Chem (Martin et al., 2002) | No |
| M2 | 45km | $40\sigma_p$ level | 57 m | Yamo | ppm | multiscale | ACM2 | SAPRC99 | Aero6 | Wesely (1989) | Henry's law | Standard[a] | Default | No |
| M3 | 45km | $40\sigma_p$ level | 57 m | Yamo | Yamo | multiscale | ACM2 | CB05 (Yarwood et al., 2005) | Aero5 | Wesely (1989) | Henry's law | Standard[a] | GEOS-Chem | No |
| M4 | 45km | $40\sigma_p$ level | 57 m | ppm | ppm | multiscale | ACM2_ inline | SAPRC99 | Aero5 | Wesely (1989) | Henry's law | Standard[a] | CHASER (Sudo et al., 2002a) | No |
| M5 | 45km | $40\sigma_p$ level | 57 m | ppm | ppm | multiscale | ACM2_ inline | SAPRC99 | Aero5 | M3DRY (Pleim et al., 2001) | Henry's law | Standard[a] | CHASER | No |
| M6 | 45km | $40\sigma_p$ level | 57 m | Yamo | Yamo | multiscale | ACM2_ inline | SAPRC99 | Aero5 | M3DRY | ACM | Standard[a] | CHASER | No |
| M7 | 45km | $40\sigma_p$ level | 29 m | Monotonic | Monotonic | no diffusion | no diffusion | RACM-ESRL with KPP ( Goliff et al.,2013 ) | MADE (Ackerman n et al., 1998) | Wesely (1989) | Henry's law | WRF/NCEP[a] | Default | No |
| M8 | 45km | $40\sigma_p$ level | 57 m | 5[th] order Monotonic | 3[th] order Monotonic | MYJ | MYJ | RACM with KPP | MADE | Wesely (1989) | AQCHEM | WRF/NCEP[a] | CHASER | Yes |
| M9 | 45km | $40\sigma_p$ level | 16 m | 5[th] order Monotonic | 3[th] order Monotonic | Smagorinsk y first order closure | YSU (Hong et al., 2006) | RADM2 (Stockwell et al., 1990) | MADE | Wesely (1989) | Easter et al., (2004) | WRF/NCEP[a] | GEOS-Chem | Yes |
| M10 | 45km | $60\sigma_p$ level | 44 m | Monotonic | 3[th] order Monotonic | 2[th] order Monotonic | YSU | RADM2 | GOCART | Wesely (1989) | Grell | WRF/ MERRA2[a] | MOZART + GOCART[b] | No |
| M11 | 45km | $20\sigma_z$ level | 50 m | Walcek and Aleksic (1998) | Walcek and Aleksic (1998) | multicale | K-theory | CBMZ (Zaveri et al.,1999) | ISORROPI A1.7 (Nenes et al.,1998) | Wesely (1989) | Henry's law | Standard[a] | CHASER | No |

| M12 | 45km | 40 $\sigma_p$ level | 54 m | Walcek and Aleksic (1998) | Walcek and Aleksic (1998) | FTCS | FTCS | SAPRC99 | Kajino et al. (2012) | Zhang et al. (2003) | Henry's law | Standard[a] | CHASER | No |
|---|---|---|---|---|---|---|---|---|---|---|---|---|---|---|
| M13 | 0.5°×0.667° | 47$\sigma_p$ level | 60 m | ppm | ppm | Lin and McElroy, 2010 | Lin and McElroy, 2010 | NO$_x$-O$_x$-HC | ISORROPIA2.0 (Fountoukis and Nenes, 2007) | Wesely | Henry's law | GEOS-5[a] | Geos-Chem | No |
| M14 | 64km | 15$\sigma_z$ level | 100 m | ppm | ppm | multiscale | ACM2 | SAPRC99 | ISORROPIA1.7 | Wesely (1989) | Henry's law | RAMS/NCEP[a] | Geos-Chem | No |


[a] Standard represents the reference meteorological field provided by MICS-Asia III project; WRF/NCEP and WRF/MERRA represents the meteorological field of the participating model itself, which was run by WRF driven by the NCEP and
Modern Era Retrospective-analysis for Research and Applications (MERRA) reanalysis dataset. RAMS/NCEP is the meteorology field run by RAMS driven by the NCEP reanalysis dataset.
[b] Boundary conditions of M10 are from MOZART and GOCART (Chin et al., 2002; Horowitz et al.,2003), which provided results for gaseous pollutants and aerosols, respectively.











**Table 2: Statistics of simulated annual mean concentrations over the NCP and PRD regions.**

| Species | Regions | Statistics | Model | | | | | | | | | | | | | |
|---|---|---|---|---|---|---|---|---|---|---|---|---|---|---|---|---|
| | | | M1 | M2 | M4 | M5 | M6 | M7 | M8 | M9 | M10 | M11 | M12 | M13 | M14 | Ense |
| NO$_2$ | NCP | R(spatial)[a] | 0.63 | 0.67 | 0.67 | 0.67 | 0.67 | 0.70 | 0.70 | 0.59 | 0.57 | 0.66 | 0.69 | - | 0.70 | 0.67 |
| | | R(temporal)[b] | 0.82 | 0.92 | 0.93 | 0.86 | 0.92 | 0.81 | 0.28 | 0.85 | 0.95 | 0.75 | 0.90 | - | 0.96 | 0.91 |
| | | MBE | -4.11 | -5.66 | -6.54 | 1.86 | -5.12 | -5.04 | 3.30 | 8.28 | -2.45 | 0.00 | -3.81 | - | -2.99 | -1.86 |
| | | NMB(%) | -17.8 | -24.5 | -28.4 | 8.0 | -22.2 | -21.9 | 14.2 | 35.9 | -10.6 | 0.02 | -16.5 | - | -13.0 | -8.0 |
| | | RMSE | 7.40 | 8.25 | 8.79 | 6.75 | 8.01 | 7.55 | 6.54 | 12.74 | 7.72 | 6.37 | 7.38 | - | 6.68 | 6.36 |
| | PRD | R(spatial)[a] | 0.12 | 0.06 | 0.07 | 0.07 | 0.06 | 0.12 | 0.20 | 0.38 | 0.00 | 0.08 | 0.12 | - | 0.02 | 0.10 |
| | | R(temporal)[b] | 0.93 | 0.80 | 0.86 | 0.88 | 0.79 | 0.68 | 0.83 | 0.95 | 0.74 | 0.74 | 0.75 | - | 0.52 | 0.86 |
| | | MBE | -6.73 | -9.84 | -7.21 | 1.96 | -6.66 | -3.99 | 3.24 | -7.61 | -1.84 | 3.02 | -5.49 | - | -5.03 | -3.85 |
| | | NMB(%) | -30.1 | -44.0 | -32.3 | 8.8 | -29.8 | -17.9 | 14.5 | -34.0 | -8.2 | 13.5 | -24.6 | - | -22.5 | -17.2 |
| | | RMSE | 11.31 | 13.14 | 12.00 | 10.80 | 11.84 | 10.60 | 8.73 | 10.69 | 10.72 | 10.51 | 11.68 | - | 12.00 | 10.15 |
| CO | NCP | R(spatial)[a] | 0.35 | 0.48 | 0.27 | 0.34 | 0.36 | 0.22 | 0.19 | 0.48 | 0.49 | 0.33 | 0.35 | -0.13 | 0.29 | 0.37 |
| | | R(temporal)[b] | 0.94 | 0.96 | 0.92 | 0.22 | 0.90 | 0.77 | 0.94 | 0.92 | 0.82 | 0.85 | 0.94 | 0.85 | 0.88 | 0.92 |
| | | MBE | -1.53 | -1.35 | -1.59 | -1.69 | -1.52 | -1.64 | -1.29 | -1.16 | -1.55 | -1.37 | -1.38 | -1.53 | -1.51 | -1.47 |
| | | NMB(%) | -68.9 | -60.9 | -71.4 | -76.2 | -68.2 | -73.7 | -58.2 | -52.0 | -70.0 | -61.6 | -62.3 | -68.9 | -68.0 | -66.2 |
| | | RMSE | 1.71 | 1.54 | 1.77 | 1.86 | 1.70 | 1.82 | 1.51 | 1.36 | 1.74 | 1.57 | 1.58 | 1.74 | 1.70 | 1.66 |
| | PRD | R(spatial)[a] | 0.04 | -0.24 | -0.25 | -0.23 | -0.22 | -0.05 | 0.08 | 0.55 | -0.02 | -0.01 | -0.22 | 0.09 | -0.21 | -0.06 |
| | | R(temporal)[b] | 0.96 | 0.91 | 0.93 | 0.84 | 0.95 | 0.90 | 0.90 | 0.96 | 0.83 | 0.87 | 0.93 | 0.76 | 0.82 | 0.94 |
| | | MBE | -0.66 | -0.64 | -0.65 | -0.64 | -0.62 | -0.64 | -0.51 | -0.57 | -0.50 | -0.51 | -0.58 | -0.52 | -0.67 | -0.59 |
| | | NMB(%) | -68.4 | -67.0 | -67.0 | -66.7 | -64.7 | -66.5 | -53.3 | -59.7 | -52.3 | -52.7 | -60.7 | -54.1 | -69.6 | -61.7 |
| | | RMSE | 0.70 | 0.70 | 0.70 | 0.69 | 0.67 | 0.69 | 0.57 | 0.62 | 0.56 | 0.57 | 0.64 | 0.58 | 0.72 | 0.65 |
| NH$_3$ | NCP | R(spatial)[a] | 0.72 | 0.70 | 0.69 | 0.70 | 0.71 | 0.65 | - | 0.70 | 0.57 | 0.62 | 0.67 | 0.61 | 0.58 | 0.69 |
| | | R(temporal)[b] | -0.48 | -0.22 | -0.45 | -0.55 | -0.41 | 0.04 | - | -0.19 | 0.64 | 0.08 | -0.37 | 0.65 | -0.04 | -0.17 |
| | | MBE | -0.69 | 2.95 | -6.14 | -6.61 | -3.89 | 4.94 | - | 21.8 | 10.5 | -0.07 | 0.31 | -5.19 | -12.2 | 0.47 |
| | | NMB(%) | -3.8 | 16.1 | -33.5 | -36.0 | -21.2 | 26.9 | - | 118.7 | 57.1 | -0.4 | 1.69 | -28.3 | -66.3 | 2.59 |
| | | RMSE | 7.20 | 10.04 | 8.95 | 9.24 | 7.48 | 8.78 | - | 29.24 | 13.48 | 8.30 | 7.33 | 8.82 | 14.48 | 7.20 |

[a] R(spatial) represents the spatial correlation coefficients between simulated and observed concentrations sampled from different stations in NCP or PRD;
[b] R(temporal) represents the temporal correlation coefficients between simulated and observed monthly mean concentrations from January to December in 2010;

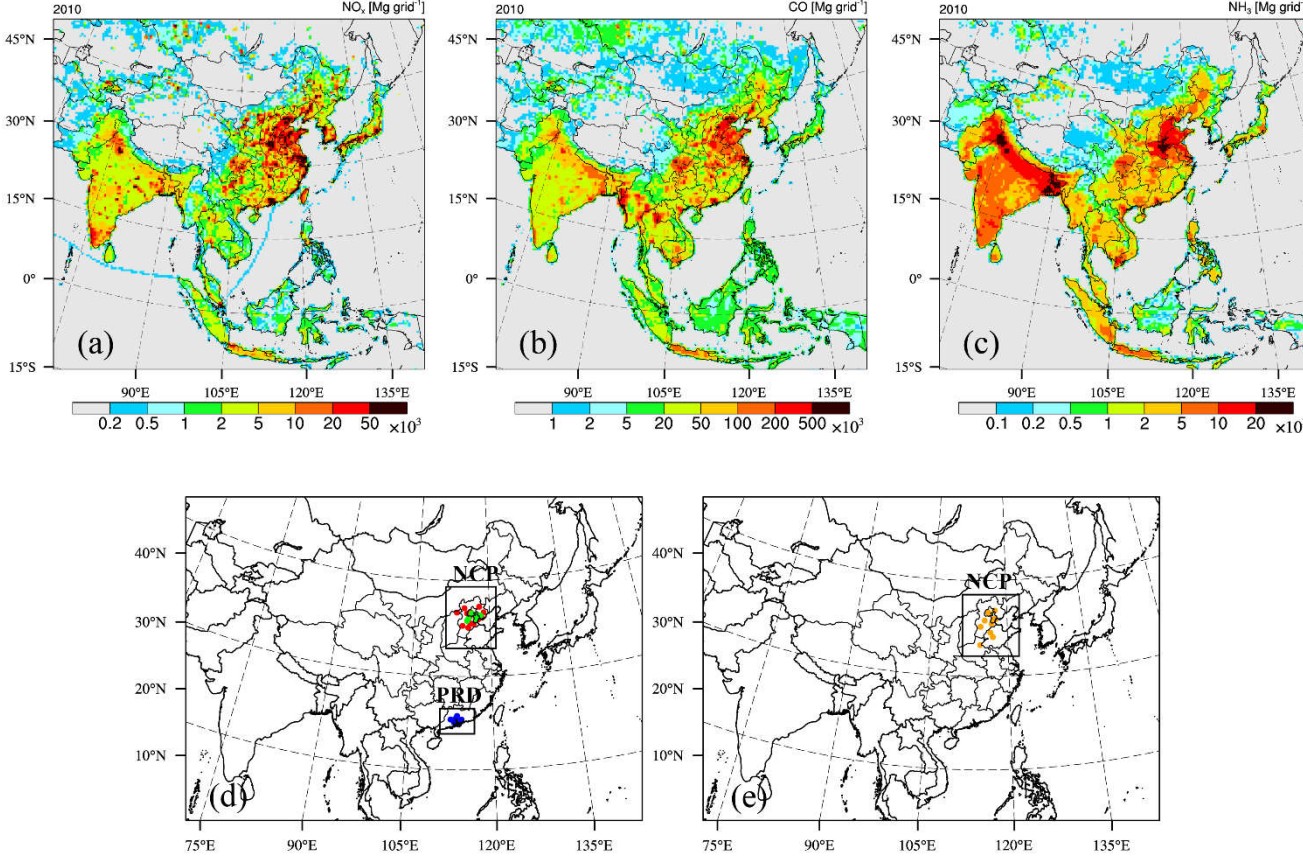


**Figure 1: Modeling domains of the participated models except M13 and M14 along with spatial distributions of the total emissions**
**of (a) NO$_x$, (b) CO and (c) NH$_3$ in 2010 provided by MICS-Asia III (upper panel), and the distributions of observation stations of (d)**
**NO$_2$ and CO over the NCP and PRD regions, as well as (e) NH$_3$ over the NCP region (lower panel). The horizontal resolution is**
**45km×45km. Note that domains of M13 and M14 are shown in fig. 7 and only six of nineteen observational sites (green) over the**
**NCP region have CO measurements.**

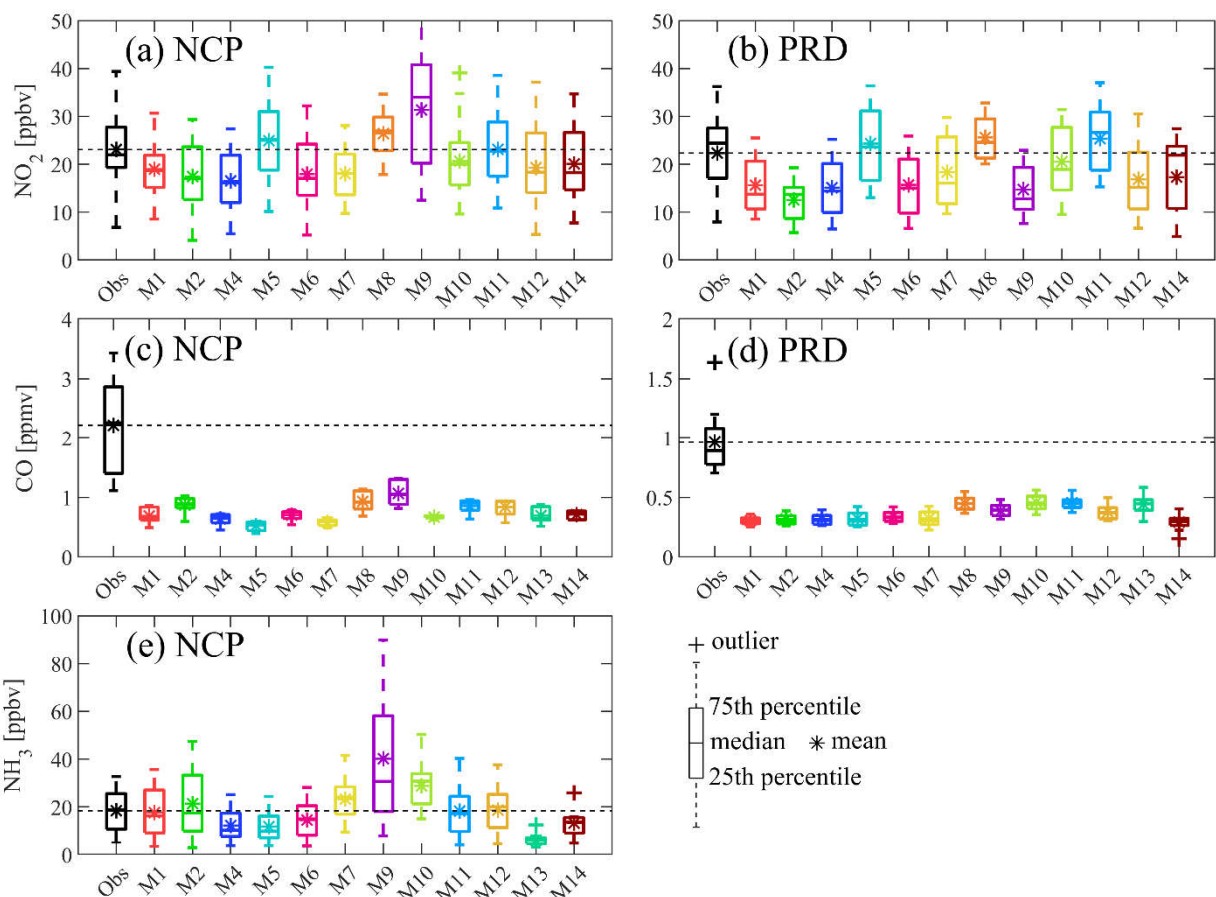


**Figure 2: Boxplot of simulated and observed annual mean NO₂, CO and NH₃ concentrations sampled from different stations over**
**the NCP (a, c, e) and PRD (b, d) regions. The outlier was defined as values larger than $q_3 + 1.5 \times (q_3 - q_1)$ or less than $q_1 -$**
**$1.5 \times (q_3 - q_1)$, where $q_3$ denotes the 75th percentile, and $q_1$ the 25th percentile. This approximately corresponds to 99.3 percent**
**coverage if the data are normally distributed.**

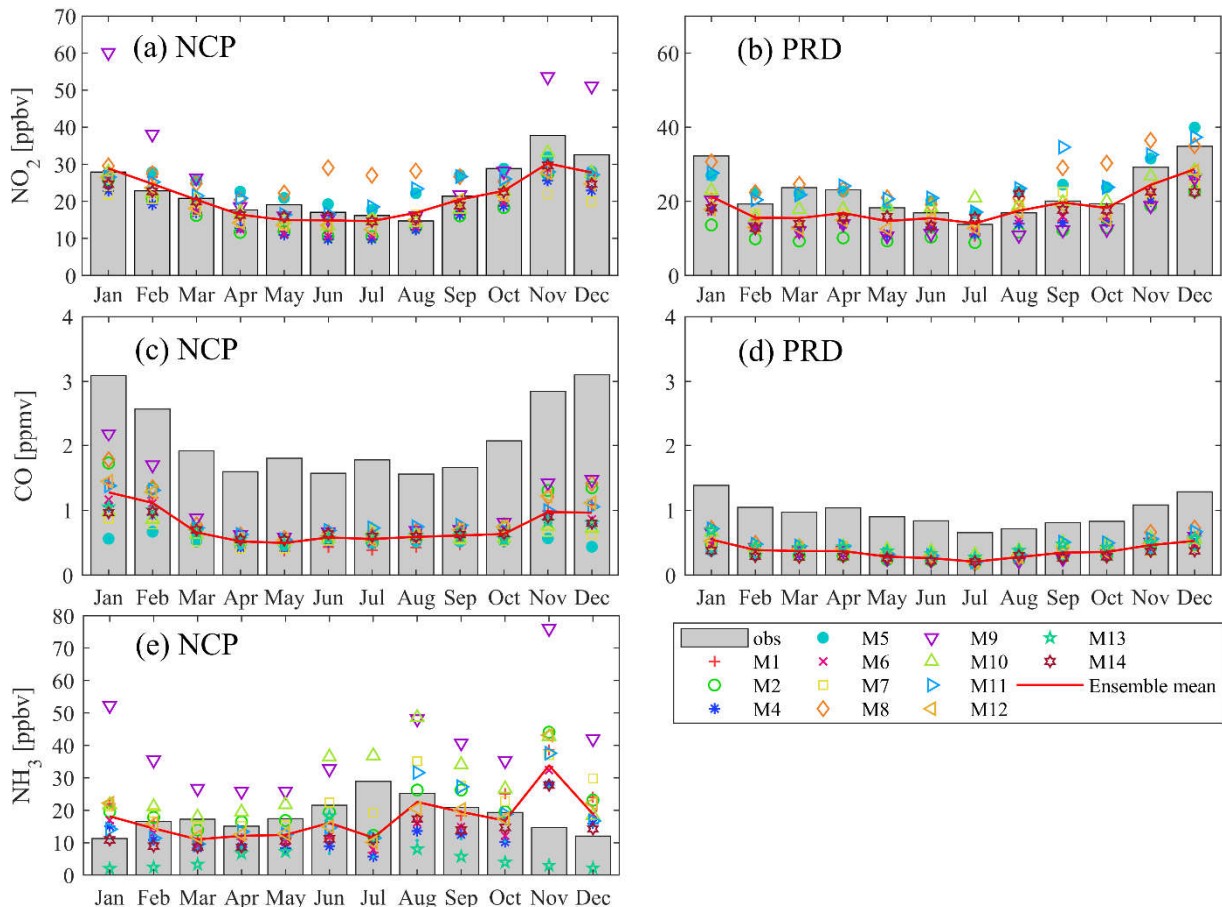


**Figure 3: Timeseries of regional mean NO₂, CO concentrations over the NCP (a, c) and PRD (b, d) regions as well as NH₃ concentrations over the NCP (e) region from January to December in year 2010.**





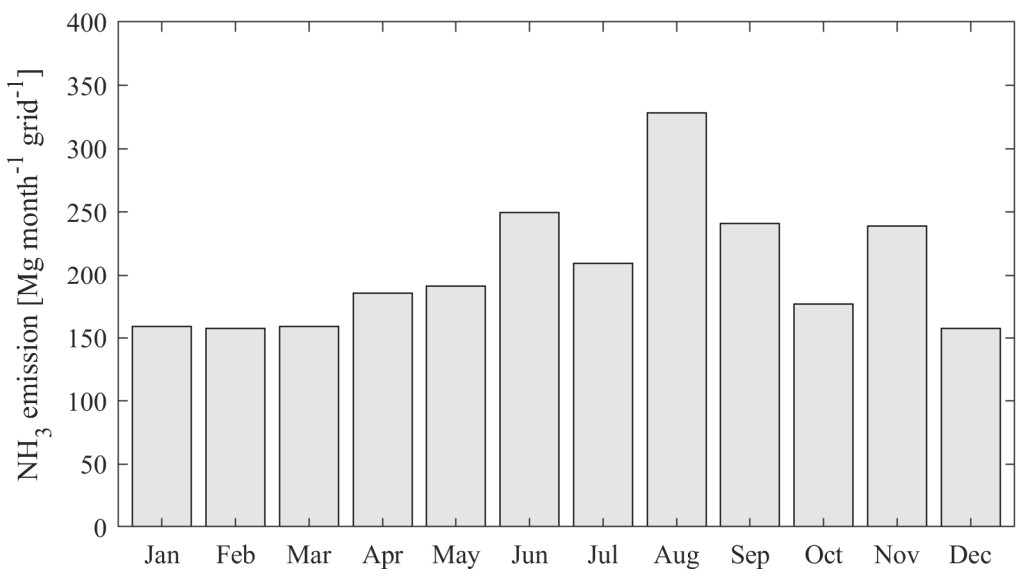


**Figure 4: Timeseries of NH₃ emissions over the NCP region provided by MICS-Asia III on a horizontal resolution of 45km×45km**
**from January to December in year 2010.**









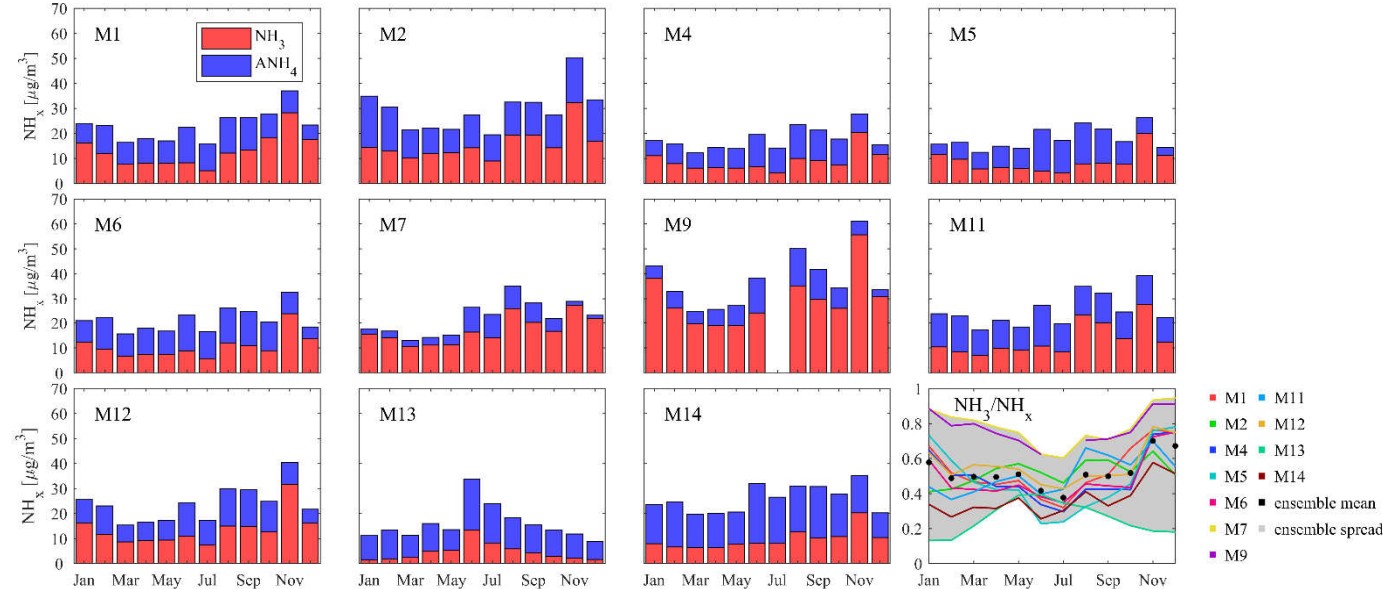


Figure 5: Timeseries of the multi-model simulated total ammonium ($NH_x = NH_3 + NH_4^+$) in atmosphere along with the ratio of gaseous $NH_3$ to the total ammonium over NCP from January to December in year 2010.












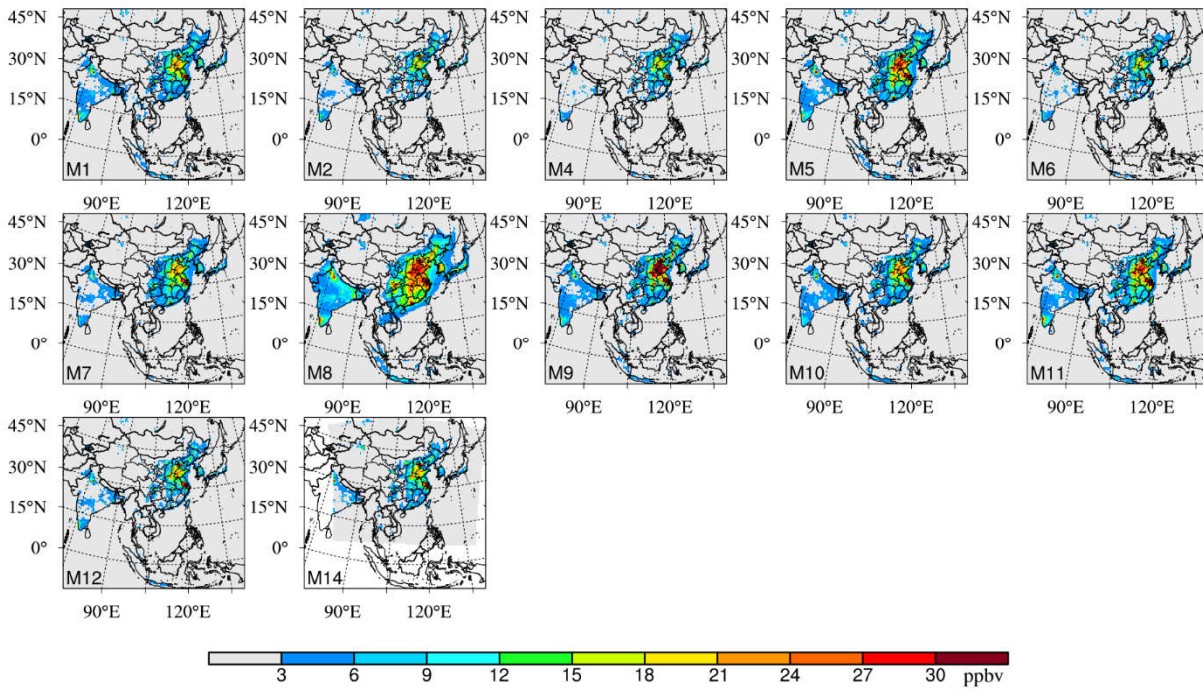


**Figure 6: Spatial distribution of the annual mean NO$_2$ concentrations from each modeling results of MICS-Asia III. Note that M13 are not included in this figure.**







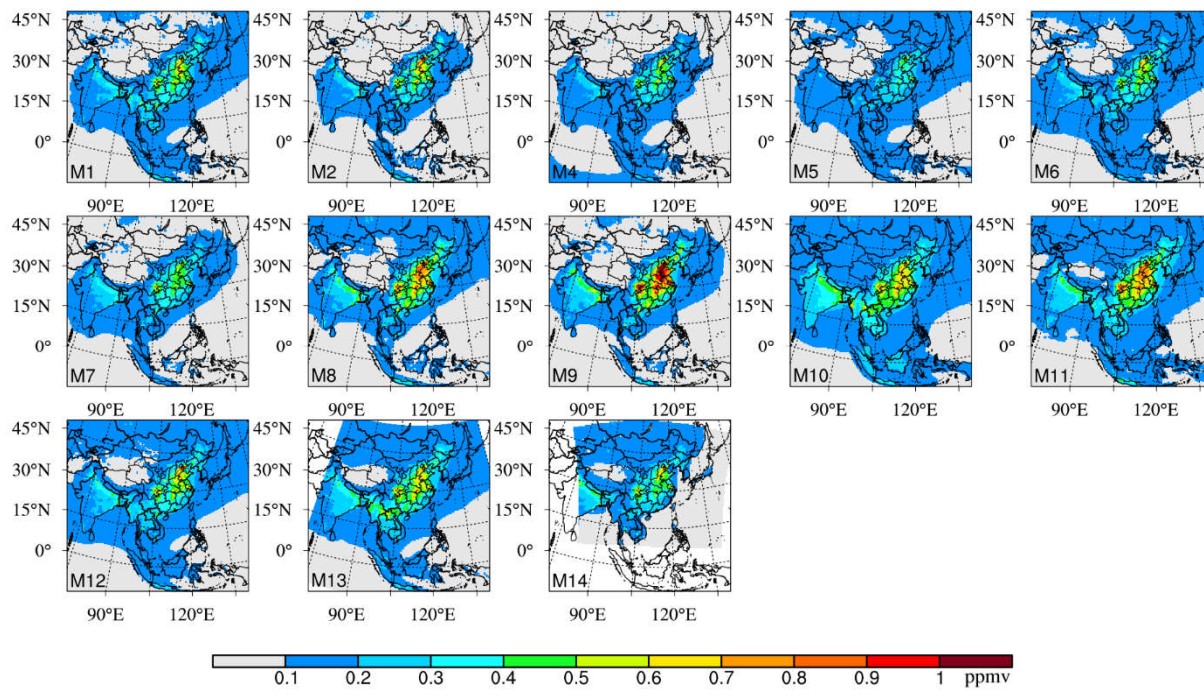


**Figure 7: Spatial distribution of the annual mean CO concentrations from each modeling results of MICS-Asia III.**







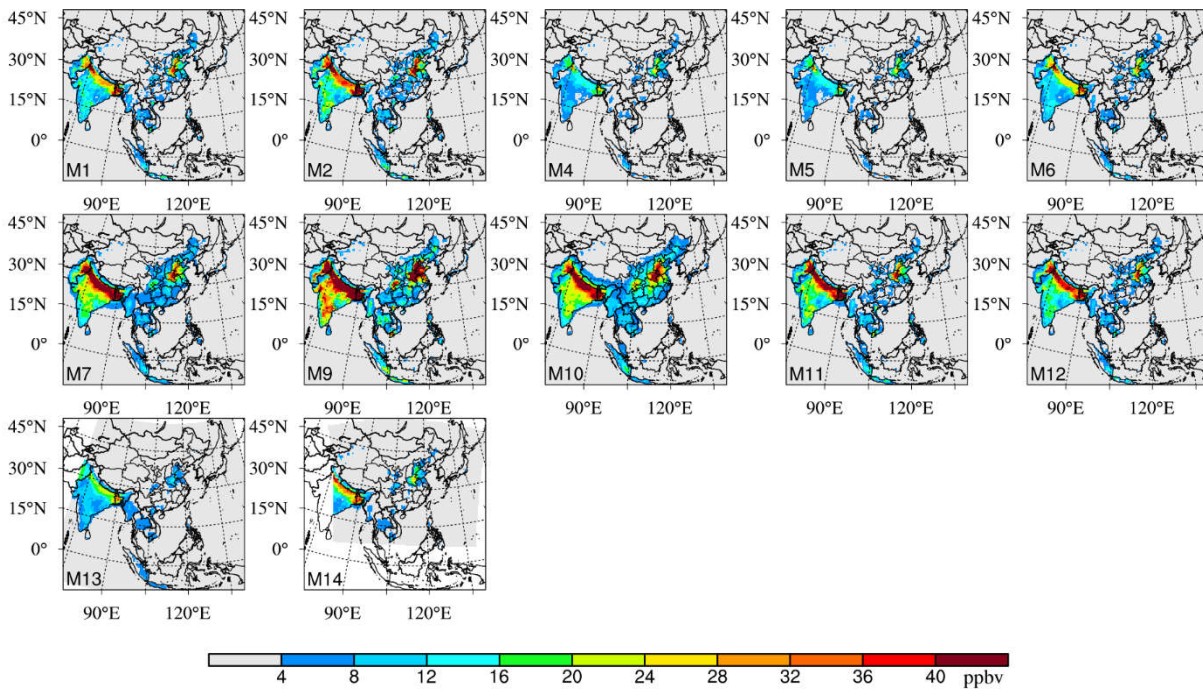


**Figure 8: Spatial distribution of the annual mean NH₃ concentrations from each modeling results of MICS-Asia III.**


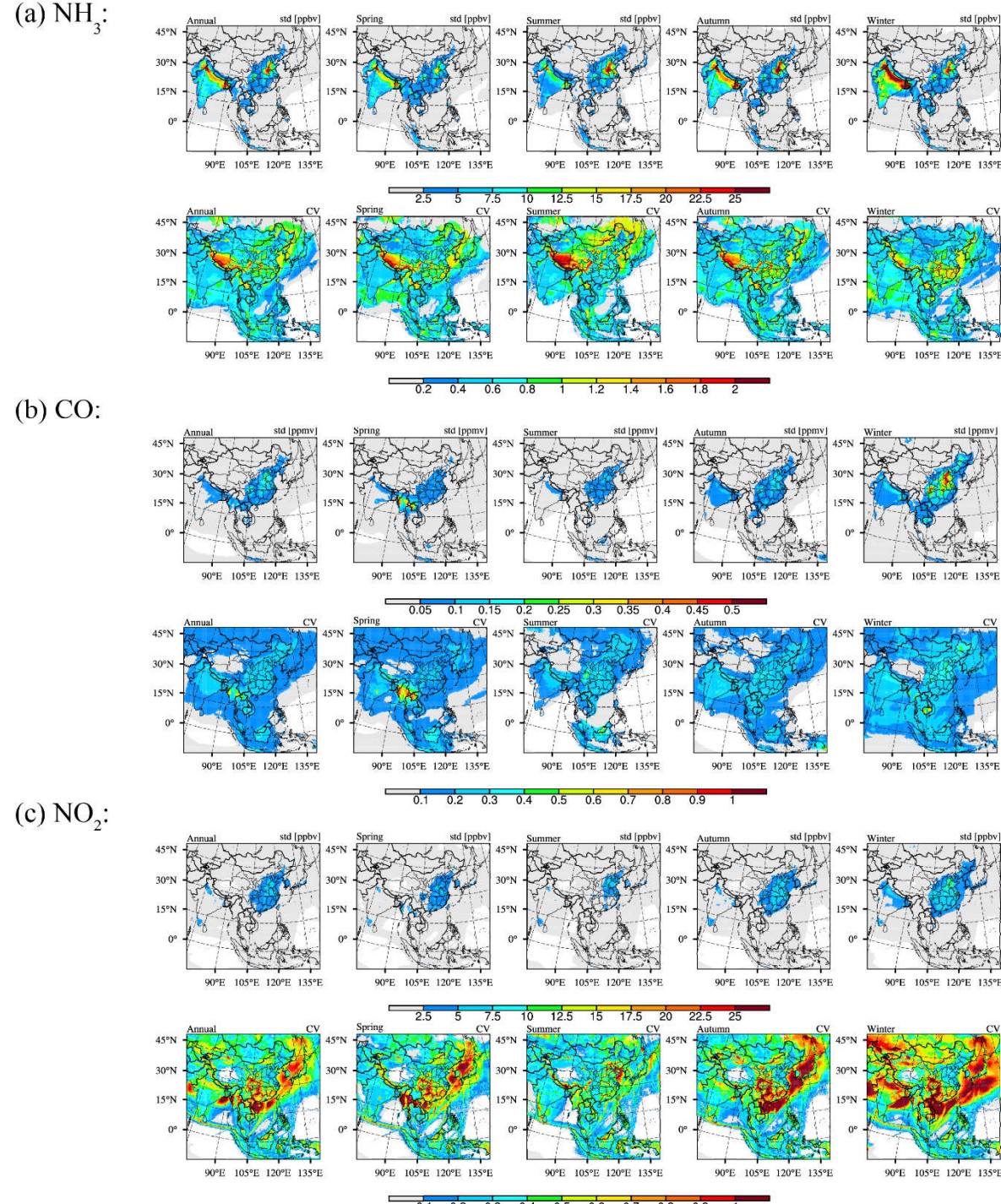


**Figure 9: Spatial distribution of the standard deviation of (a) NH₃, (b) CO and (c) NO₂ multi-model predictions in MICS-Asia III, as well as the corresponding distribution of CV on the annual and seasonal basis.**