# Peer review of "Evaluation and uncertainty investigation of the NO2, CO and NH3"

_Atmospheric Chemistry and Physics, 2018_

## Referee Comment (RC1) · Anonymous Referee #2 · 3 Jun 2019

This paper conducts ensemble air quality modeling of NO2, CO, and NH3 over Asia, and evaluates model performance using measurements data in the North China Plain and Pearl River Delta regions. 14 models including 13 regional models and one global model with common emission inventory, meteorological fields, modeling domain, and horizontal resolutions were used for the ensemble modeling. The results show that NO2 and CO simulations are mostly underestimated and NH3 modeling mismatches the observed temporal variations. Possible reasons for the model structural uncertainties and recommendations for the future studies are given by the authors. This paper is good in general and within the scope of Atmospheric Chemistry and Physics. I recommend for publication once the concerns expressed below are addressed. Some

specific comments: 1. Although 14 models are required to use standard meteorological field, the configurations of meteorological models may not be identical. The author also needs to list the configurations of each meteorological model as in Table 1. Meanwhile, since the meteorological parameters have large impact on the modeled concentrations, the modeled meteorological fields also need to be validated against observed data. 2. The model performance in PRD is much worse than that in NCP. The author concludes that it is because of coarse horizontal resolution. I think uncertainties may primarily come from the emission inventory, especially spatial allocations from different emission sectors are not well resolved in the PRD region. I suggest the author use one or two models with finer resolution to test the model performance again in PRD, to see if the horizontal resolution is the main problem as the author demonstrated. 3. I agreed with the author using the available NH3 observations from the other years as an alternative to evaluate the performance of different models. However, to evaluate the modeled temporal variations using observed data from different years may not be appropriate, because the NH3 emissions vary year by year, and control measures may be applied in year of measurement conducted. 4. Figure 5 is an interesting finding in this paper. I am surprised that the NH3 gas-aerosol partitioning simulations from different models have such large discrepancies. Is it because the chemical mechanisms in different models treating NH3 different? Otherwise, please explain why does such large discrepancy of NH3 gas-aerosol partitioning occur in different models. 5. In summary, the author makes a few recommendations for future studies. I think inversions of NOx and CO emissions will help to reduce uncertainties in emission inventory and improve model performance, since many inverse modeling works of NOx and CO emissions have been done using satellite as well as ground observations. However, I have doubts on inversion of NH3 because of the reactivity and uncertainties in the chemical pathways of NH3 gas. 6. In page 1, line 40, change "peral"to"pearl". 7. In page 4, line 4, missing "plain"; line 5, change "peral"to"pearl". 8. In Figure 1, I think the color of CO measurement sites in NCP should be"green"instead of "blue".

---

## Referee Comment (RC2) · Anonymous Referee #3 · 14 Sep 2019

This work evaluated 14 model simulations of NO2, CO and NH3 over China under the framework of MICS-Asia III with the aim to assess the capability and uncertainty of current CTMs in East Asia. Model results were provided by a larger number of independent groups and covered a full year (2010). The results show that most models well captured the monthly and spatial patterns of NO2 in NCP though NO2 levels are slightly underestimated, but relatively poor model performance was observed in the PRD region. All models significantly underpredict CO concentrations both in the NCP and PRD regions and failed to reproduce the observed monthly variation of NH3 in NCP.

[Figure]

This work quantifies the impacts of model uncertainties on simulations of the three primary gases, which shows the large uncertainty (spread) in simulating more reactive and/or short-lived primary pollutants (e.g. NH3). This work is important and valuable to the scientific and regulatory community as it provides information on the capability and limitations of some widely used models. The manuscript is well organized and well written, and model results (tables and figures) are clearly presented. I recommend its publication after the authors have addressed my comments listed below.

1. For comparison with the NO2 measured from the regular monitoring networks, please note that these networks employ a thermal conversion method which converts NO2 to NO, followed by detection of NO. This method is known to overestimate NO2 as it also converts other NOy species such as HONO and PAN etc (e.g., Xu et al., 2013). It is important to correct this measurement problem before making the comparison, using, for example, the approach by Zhang et a. (2017). After corrections of the measurement data, a closer agreement would be seen between the modelled results and the observations in the present work. If the author cannot make such corrections in view of a large number of groups involved, at least some discussions should be provided on this point. References

Xu, Z., T. Wang, L. K. Xue, P. K. K. Louie, C. W. Y. Luk, J. Gao, S. L. Wang, F. H. Chai, and W. X. Wang. "Evaluating the Uncertainties of Thermal Catalytic Conversion in Measuring Atmospheric Nitrogen Dioxide at Four Differently Polluted Sites in China." Atmospheric Environment 76 (Sep 2013): 221-26. http://dx.doi.org/10.1016/j.atmosenv.2012.09.043.

Zhang, L., Q. Y. Li, T. Wang, R. Ahmadov, Q. Zhang, M. Li, and M. Y. Lv. "Combined Impacts of Nitrous Acid and Nitryl Chloride on Lower-Tropospheric Ozone: New Module Development in WRF-Chem and Application to China." Atmospheric Chemistry and Physics 17, no. 16 (Aug 2017): 9733-50. http://dx.doi.org/10.5194/acp-17-9733-2017.

2. Section 2.2. The comparison of NO2 and CO concentrations are only for NCP and

[Figure]

PRD. Any reasons why not to include other regions?

3. For simulations of NO2 (and NH3), accurate representation of nitrogen chemistry is critical. Recent studies have shown that the HONO sources may be under-represented in some models which would give rise to larger simulated NO2 values (as it underestimates the oxidation of NO2 by OH) (e.g., Zhang et al., 2017; Fu et al., 2019); N2O5 uptake on aerosol may be treated differently in models which could also affect the NO2 simulations. Therefore, in discussing the discrepancy in modelled NO2, information on how models treat these nitrogen processes would be helpful.

Fu, Xiao; Wang, Tao; Zhang, Li; Li, Qinyi; Wang, Zhe; Xia, Men; Yun, Hui; Wang, Weihao; Yu, Chuan; Yue, Dingli; Zhou, Yan; Zheng, Junyun; Han, Rui "The significant contribution of HONO to secondary pollutants during a severe winter pollution event in southern China" ATMOSPHERIC CHEMISTRY AND PHYSICS, Volume: 19, Issue: 1, Pages: 1-14, JAN 2019

4. The photo-chemical mechanisms used in this study are CBMZ, CB05, and SAPRC 99, and some of them have an updated version such as CB06 and SPARC 07. These updated mechanisms could give different results on model performance. The author is advised to discuss this point to alert the reader that their conclusion may not be applicable to the newer version of the respective mechanism.

5. The present comparisons focused on yearly and monthly model performance. It would be interesting to show how different models compare during severe pollution episodes. An important application of CTMs in China is to forecast severe episodes based on which emergency source control measures are activated.

6. The model comparisons were conducted for NO2, CO, and NH3. How about SO2, which is another important primary pollutant? I think the reader would be interested in seeing the model performance for SO2 as well.

7. Conclusion (1) recommends to improve the CO emission inventory which is for year

2010. Does the recent CO emission have similar problem?

8. This study reveals a large spread of model simulations for reactive gases. As the exact causes for the difference have not been identified for the individual model, I think it is important to emphasize the need to validate the individual model before using its results to make important policy recommendation.

minor comments:

Line 40 page1, line 4 page 4, the "Peral" should be "Pearl".

---

## Author Comment (AC1) · 25 Oct 2019

The comment was uploaded in the form of a supplement:
https://www.atmos-chem-phys-discuss.net/acp-2018-1158/acp-2018-1158-AC1-supplement.pdf

---

## Author Response (AR1)

**Response to Referee #2 (acp-2018-1158)**

We Thank Reviewer for his/her constructive comments

Responses to the Specific comments

**General comments:** This paper conducts ensemble air quality modeling of $NO_2$, CO, and $NH_3$ over Asia, and evaluates model performance using measurements data in the North China Plain and Pearl River Delta regions. 14 models including 13 regional models and one global model with common emission inventory, meteorological fields, modeling domain, and horizontal resolutions were used for the ensemble modeling. The results show that $NO_2$ and CO simulations are mostly underestimated and $NH_3$ modeling mismatches the observed temporal variations. Possible reasons for the model structural uncertainties and recommendations for the future studies are given by the authors. This paper is good in general and within the scope of Atmospheric Chemistry and Physics. I recommend for publication once the concerns expressed below are addressed.

**Reply:** The authors appreciate the reviewer for his/her constructive and up-to-point comments. We have carefully considered the comments and revised the manuscript accordingly. Please refer to our responses for more details given below.

**Comment 1:** Although 14 models are required to use standard meteorological field, the configurations of meteorological models may not be identical. The author also needs to list the configurations of each meteorological model as in Table 1. Meanwhile, since the meteorological parameters have large impact on the modeled concentrations, the modeled meteorological fields also need to be validated against observed data.

**Reply:** Thanks for this important comment. In MICS-Asia III, most of the CTMs used the standard meteorological fields simulated by WRFv3.4.1, except the WRF-Chem models (M7-M10), GEOS-Chem (M13) and RAMS-CMAQ (M14) which used their own meteorological fields. Following the reviewer's suggestion, a new table listing the configurations of the meteorological simulations were added to the supplementary material (*please see table S1 in the supplementary*). Table R1 presents the configurations of the standard meteorological simulation as well as those used in WRF-Chem models (M7–M10). The GEOS-Chem (M13) was driven by the GEOS-5 assimilated meteorological fields from the Goddard Earth Observing System of NASA Global Modeling Assimilation Office, and the RAMS-CMAQ (M14) was driven by the Regional Atmospheric Modeling System (RAMS). For WRF-Chem models, the configurations of their meteorological models were only slightly different from the standard model (Table R1). For example, M7 used the same parametrization schemes as the standard model in terms of the microphysics, radiation, boundary layer, cumulus physics and surface physics. The other WRF-Chem models differed from the standard model only in one or two parametrization schemes.

Table R1: Meteorological configurations for the standard meteorological field and different WRF-Chem models

| No | Microphysics | Longwave radiation | shortwave radiation | Boundary layer | Cumulus physics | surface physics |
|---|---|---|---|---|---|---|
| Standard | Lin et al. scheme | RRTMG scheme | Goddard shortwave scheme | YSU scheme | Grell 3D ensemble scheme | Unified Noah land-surface model |
| M7 | Lin et al. scheme | RRTM scheme | Goddard shortwave | YSU scheme | Grell 3D ensemble scheme | Unified Noah land-surface model |
| M8 | Lin et al. scheme | RRTMG scheme | RRTMG scheme | Mellor-Yamada-Janjic TKE scheme | Grell 3D ensemble scheme | Unified Noah land-surface model |
| M9 | Lin et al. scheme | RRTMG scheme | RRTMG scheme | YSU scheme | Grell 3D ensemble scheme | Unified Noah land-surface model |
| M10 | Goddard Cumulus Ensemble | Goddard longwave scheme | Goddard shortwave scheme | YSU scheme | Grell 3D ensemble scheme | Unified Noah land-surface model |

We agree with the reviewer that the meteorological parameters have large impacts on the simulations of atmospheric chemistry. As suggested, we have added the evaluations of the wind speed (u-wind and v-wind), relative humidity (RH) and air temperature (T) simulated by the standard meteorological model in the revised manuscript (*please see lines 139–143 in the revised manuscript and Sect.S1 in the supplementary*). These parameters are all important meteorological factors that influences the simulations of $NO_2$, CO and $NH_3$ concentrations. For example, the wind speed determines the transport of species and the air temperature influences the reaction rates of thermal chemical reactions. The relative humidity and temperature also have impacts on the thermodynamic equilibrium of gases and aerosols.

Three-hourly meteorological observations from the Integrated Surface Database (ISD) compiled by the National Oceanic and Atmospheric Administration (NOAA), U.S. (Smith et al., 2011) were used in this study. We focused on the evaluations of meteorological simulations over the North China Plain (NCP) and the Pearl River Delta region with the observation sites used in evaluation shown in Fig.R1. Figure R2 shows the averaged time series of simulated meteorological parameters and observations over the NCP region from January, 2010 to December, 2010 with an interval of three hours. The evaluation statistics, including correlation coefficient (R), mean bias error (MBE) and root of mean square error (RMSE), were summarized in Table R2. It clearly shows that the standard meteorology simulations well captured the main features of the observed meteorological conditions in the NCP region throughout the year with high correlation coefficient, small biases and low RMSE for all meteorological parameters. Similar results could be obtained from the evaluations of meteorological conditions over the PRD region (fig R3). These results suggested that the standard meteorological simulations can well reproduce the meteorological conditions of the NCP and PRD region.

Table R2: Evaluation metrics of the standard meteorological simulation

|  | NCP | | | PRD | | |
|---|---|---|---|---|---|---|
|  | R | MBE | RMSE | R | MBE | RMSE |
| temp (℃) | 1.00 | 0.21 | 1.08 | 1.00 | -0.22 | 0.71 |
| RH (%) | 0.97 | -0.16 | 5.15 | 0.97 | 3.42 | 4.82 |
| u-wind (m/s) | 0.91 | -0.08 | 0.63 | 0.82 | -0.20 | 0.53 |
| v-wind (m/s) | 0.93 | 0.33 | 0.76 | 0.93 | 0.05 | 0.81 |

[Figure]

Figure R1: Spatial distributions of the meteorological observation sites from the ISD over the NCP region (left panel) and the PRD region (right panel).

[Figure]

Figure R2: Time series of the simulated and observed meteorological parameters over the NCP region form January 2010 to December 2010 with an interval of three hours.

[Figure]

Figure R3: Same as Figure R2 but for the PRD region.

**Changes in the manuscript: lines 139–143.**

**Changes in the supplementary: Sect.S1, Table S1-2 and Figure S1-3.**

**Comment 2:** The model performance in PRD is much worse than that in NCP. The author concludes that it is because of coarse horizontal resolution. I think uncertainties may primarily come from the emission inventory, especially spatial allocations from different emission sectors are not well resolved in the PRD region. I suggest the author use one or two models with finer resolution to test the model performance again in PRD, to see if the horizontal resolution is the main problem as the author demonstrated.

**Reply:** Thanks for this valuable suggestion. As suggested, a full-year run of NAQPMS model with finer horizontal resolutions has been conducted to investigate the impacts of horizontal resolutions on the simulations of $NO_2$ and CO over the PRD region. The NAQPMS model is one of the participating CTMs in MICS-Asia III. Two nested domains with finer horizontal resolutions were added to the original modeling domain of MICS-Asia III, which are shown in Fig. R4. The first domain (D1) is identical to the modeling domain of MICS-Asia III with horizontal resolution of 45km. The second domain (D2) covers most part of southeast China with horizontal resolution of 15km; the third domain has the finest horizontal resolution (5km) covering the PRD region and its surrounding areas. The chemical configurations of NAQPMS in each modeling domain were completely identical to those used in MICS-Asia III. Meteorological fields for each modeling domain were simulated by the WRF model version 3.4.1, same as the standard meteorological model in MICS-Asia III. The WRF configurations were also the same as those used in the standard meteorological simulations except two additional nested domains were added (Fig. R4). The emission inventories and boundary conditions in D1 were provided by the standard input datasets of MICS-Asia III. Since MICS-Asia III only provided the emission inventories and boundary conditions at 45km horizontal resolution, in D2 and D3, the emission rates ($\mu g/m^2/s$) and boundary conditions over one model grid were simply obtained from the corresponding model grid in its parent domain. This means that although we used the finer horizontal resolutions in D2 and D3, the resolutions of emission inventories and boundary conditions in D2 and D3 were the same as those used in D1. Therefore, the horizontal resolutions were only dynamically increased in D2 and D3. The simulation results from different modeling domains were then compared with each other to investigate the dynamical impacts of horizontal resolution on the model performance.

[Figure]

Figure R4: Modeling domain of the sensitivity experiment with different horizontal resolutions.

Figure R5 shows the spatial distributions of the observed annual mean $NO_2$ concentrations in the PRD region overlay the simulation results using different horizontal resolutions. We can clearly see that the coarse modeling results (D1) cannot resolve the high spatial variability of $NO_2$ concentrations in the PRD region, which is consistent with what we found from the MICS-Asia III. For simulations using finer horizontal resolutions (D2 and D3), although the spatial scales of $NO_2$ observations can be resolved by the 15km and 5km resolutions, the modeling results still show poor performance in capturing the observed spatial variability of $NO_2$ concentrations, with calculated correlation coefficient only of 0.03 and 0.02, respectively (table R2), even worse than the coarse resolutions. Similar results could be obtained from the comparisons of CO observations and simulations with different horizontal resolutions (Fig.R6). These results indicated that the poor model performance in the PRD region may not be attributed to the resolution of model but more related to the resolution and/or spatial allocation of the emission inventories in the PRD region. These results also suggested that only increasing the resolution of the model may not help improve the model performance.

Thus, as the reviewer suggested, the poor model performance in PRD may be more related to coarse resolution and/or inappropriate spatial allocation of the emission inventories in PRD region. Based on these results, we have revised the abstract (*please see lines 43–45 in the revised manuscript*), Section 3.3.1 (*please see lines 244–254 in the revised manuscript*) and Summary (*please see lines 420–424 in the revised manuscript*) part of the manuscript. Analysis of this sensitivity experiments were also added to the supplementary material (*please see Sect.S3 in the supplementary material*).

Table R2: Table S3: Evaluation metrics of the simulated annual mean NO$_2$ and CO concentrations over the PRD region with different horizontal resolutions.

| | NO$_2$ (ppbv) | | | | CO (ppmv) | | | |
|---|---|---|---|---|---|---|---|---|
| | Spatial R | MBE | NMB (%) | RMSE | Spatial R | MBE | MBE (%) | RMSE |
| 45km | 0.09 | 2.99 | 13.37 | 10.53 | 0.00 | -0.51 | -52.85 | 0.57 |
| 15km | 0.03 | 2.19 | 9.81 | 10.15 | 0.00 | -0.54 | -56.25 | 0.60 |
| 5km | 0.02 | 0.58 | 2.59 | 10.23 | -0.10 | -0.58 | -59.23 | 0.62 |

[Figure]

Figure R5: Spatial distributions of the observed and multi-resolution simulated annual mean NO$_2$ concentrations over the PRD region.

[Figure]

Figure R6: Same as figure R5 but for CO concentrations.

**Changes in the manuscript: lines 43–45, lines 244-254 and lines 420-424.**

**Changes in the supplementary: Sect.S3 and Figure S4-6.**

**Comment 3**: I agreed with the author using the available NH₃ observations from the other years as an alternative to evaluate the performance of different models. However, to evaluate the modeled temporal variations using observed data from different years may not be appropriate, because the NH₃ emissions vary year by year, and control measures may be applied in year of measurement conducted.

**Reply:** Thanks for this comment. We agree with the reviewer that the use of NH₃ observations from different years may be inappropriate for evaluating the modeled temporal variations due to the emission changes of NH₃. In the revised manuscript, this problem has been discussed using the satellite retrievals of NH₃ total columns from IASI (Infrared Atmospheric Sounding Interferometer) since we did not obtain the direct surface observations of NH₃ concentration over China in 2010 (*please see lines 202–207 in the revised manuscript*). We used the ANNI-NH3-v2.1R-I retrieval product(Van Damme et al., 2017;Van Damme et al., 2018) in this study which is the reanalysis version of NH₃ retrievals from IASI instruments and provides the daily morning (~9:30 am local time) NH₃ total columns from year 2008 to 2016. The morning orbit was used since IASI is generally more sensitive to the atmospheric boundary layer at this time due to more favorable thermal conditions, which could provide more information on the NH$_3$ concentrations in the boundary layer where NH$_3$ is emitted. This dataset was produced by Van Damme et al., 2018 based on the conversion of hyperspectral range indices (HRIs) using an Artificial Neural Network(Whitburn et al., 2016). It uses the ERA-interim ECWMF meteorological input data rather than the operationally provided EUMETSAT IASI Level 2 (L2) data used for the standard near-real-time version, which is more coherent in time and suitable for the study of temporal variations.

To facilitate comparisons, the NH$_3$ total columns were averaged to the monthly data at 45km × 45km MICS-Asia grids. A comparison of surface NH$_3$ observation from AMoN-China and NH$_3$ total columns from IASI was first conducted to see if IASI measurement could reasonably represent the monthly variations of surface NH$_3$ concentrations, which is shown in Fig.R7. We can see that the IASI measurement can generally well represent the monthly variations of surface NH$_3$ concentrations over the NCP region. Both two datasets show a very strong summer peak in July and a subpeak in Spring. However, the IASI NH$_3$ columns show a steeper monthly variations than the surface NH$_3$ observations suggested. The month of the subpeak in spring is also different between these two datasets. Nevertheless, the IASI measurement well captured the major monthly patterns of the surface NH$_3$ concentrations, which can be used to qualitatively evaluate the modeled monthly variations.

Figure R8 shows the spatial distributions of the monthly mean IASI NH$_3$ total columns over the modeling domain of MICS-Asia III in year 2010. The IASI measurement has a good agreement with the modeled results regarding the spatial distributions of the NH$_3$ concentrations over East Asia with high columns over Indo-Gangetic Plain and the North China Plain (NCP). However, large discrepancy exists in the monthly variations of NH$_3$ concentrations over the NCP region between model results and IASI measurements. Consistent with Fig. R7, The IASI NH$_3$ total columns exhibit significant monthly variations over the NCP region with a strong summer peak in July while the model results shows peak values in November (*Fig.3e in the revised manuscript*). This is consistent with the comparisons of surface NH$_3$ concentrations, which further confirms the potential deficiency of current CTMs in reproducing the monthly variations of NH$_3$ concentrations over NCP.

We also plotted the time series of monthly IASI NH$_3$ total columns averaged over NCP from January, 2008 to December, 2016 to investigate the interannual change of the monthly variations of NH$_3$ concentrations over NCP, which is shown in Fig. R8. We can see that although there are some interannual changes of magnitude of NH$_3$ total columns, the monthly pattern of NH$_3$ total columns is quite similar among different years, which suggests that the interannual change of monthly variation of NH$_3$

concentrations were very small in these years. Thus, the NH₃ observations from different years could still provide us valuable information on the monthly variation of NH₃ concentrations, which can be used as an alternative to qualitatively evaluate the modeled monthly variation.

These results have been summarized in the revised manuscript (*please see lines 312–323 in the revised manuscript*) and the supplementary (*please see Figure S7-8 in supplementary*)

[Figure]

Figure R7: Time series of the surface NH₃ concentrations from AMoN-China (left panel) and NH₃ total columns from IASI (right panel) over the NCP region during September 2015 – August 2016. Note that we reordered the months to better characterize the monthly variations

[Figure]

Figure R8: Spatial distributions of the monthly mean IASI NH₃ total columns over the modeling domain of MICS-Asia III

[Figure]

Figure R9: Monthly series of IASI measured NH₃ total columns over the NCP region from year 2008 to 2016.

**Changes in the manuscript: lines 202–207 and lines 312-323.**

**Changes in the supplementary: Sect.S2 and Figure S7-8.**

**Comment 4:** Figure 5 is an interesting finding in this paper. I am surprised that the NH₃ gas-aerosol partitioning simulations from different models have such large discrepancies. Is it because the chemical mechanisms in different models treating NH₃ different? Otherwise, please explain why does such large discrepancy of NH3 gas-aerosol partitioning occur in different models.

**Reply:** Thanks for this comment. As the review mentioned, the gas-chemistry mechanism may contribute to the differences in the modeled gas-aerosol partitioning of NH₃. M9 used the RADM2 mechanism which give lower reaction rates of oxidation of $SO_2$ and $NO_2$ by OH radical as compiled by Tan et al., 2019, leading to lower productions of acid and thus lower conversion rate of NH₃ to $NH_4^+$. Besides, the hydrolysis of $N_2O_5$ was not considered in M7, which leads to a lower tendency in the prediction of $NO_3^-$ (Chen et al., 2019), and partly explains the higher NH₃ predictions of M7. On the contrary, M14 showed a much lower $NH_3/NH_x$ ratio than most models, which would be related to its higher production rates of sulfate than other models (Chen et al., 2019). For M10, the higher NH₃ predictions of M10 would be related to the inorganic aerosol module used in the model (GOCART). The GOCART aerosol module did not consider the $NH_4^+$ aerosol, thus the emitted $NH_3$ would be only presented as the gas phase in the atmosphere, leading to higher $NH_3$ predictions in M10. This may also help explain the different monthly variations of $NH_3$ concentrations seen in M10. Without the considerations of $NH_4^+$, the monthly variations of $NH_3$ concentrations in M10 were more consistent with the monthly variations of $NH_3$ emissions. This again highlighted the importance of gas-aerosol partitioning of $NH_3$ on the predictions of monthly variations of $NH_3$ concentrations.

Based on these results, we have added more discussions on the potential reasons for the differences in the modeled gas-aerosol partitioning of $NH_3$ in the revised manuscript (*please see lines 335–339 and lines 343–349 in the revised manuscript*).

**Changes in the manuscript: lines 335–339 and lines 343-349;**

**Comment 5**: In summary, the author makes a few recommendations for future studies. I think inversions of NOx and CO emissions will help to reduce uncertainties in emission inventory and improve model performance, since many inverse modeling works of NOx and CO emissions have been done using satellite as well as ground observations. However, I have doubts on inversion of NH3 because of the reactivity and uncertainties in the chemical pathways of $NH_3$ gas.

**Reply:** Thanks for this comment. We agree with the reviewer that the inversion of $NH_3$ emissions (top-down method) would be more complicated than that for the $NO_x$ and CO emissions due to the larger uncertainties in modeling the atmospheric processes of $NH_3$. However, the inversion of $NH_3$ emissions could still provide valuable clues for verifying bottom-up emission inventories (Zhang et al., 2009) if the models were well validated. In addition, Most of $NH_3$ is emitted from the non-point sources like livestock or fertilizer uses, which is difficult to be measured over a large domain. As a result, detailed activity data and emission factors for $NH_3$ emissions are rarely available nationally, leading to high uncertainties in the spatial and temporal patterns of $NH_3$ emissions. Using the ground or satellite measurements, the top-down methods could give valuable information on the spatial and temporal characteristics of $NH_3$ emission inventories (Li et al., 2017). Therefore, although there are uncertainties in modeling the processes of $NH_3$, several inversion studies has been conducted for $NH_3$ emissions in U.S., Europe and also China (Gilliland et al., 2003;Paulot et al., 2014;Zhu et al., 2013;Zhang et al., 2018), which has provided valuable suggestions to the improvement of $NH_3$ emission inventories. Thus, we still believe the top-down methods could help improve the development of $NH_3$ emissions, however, we have clarified the needs of model validation before the inversion of $NH_3$ emissions in the revised manuscript (*please see lines 454–461 in*

*the revised manuscript*), which as follows:

*"The inversion of NH₃ emissions would be more complicated than the inversion of CO emissions due to the larger uncertainties in modeling the atmospheric processes of NH₃. Nevertheless, it could still provide valuable clues for verifying the bottom-up emission inventories (Zhang et al., 2009) if the models were well validated. In addition, by using the ground or satellite measurements, the top-down methods could also give valuable information on the spatial and temporal patterns of NH₃ emissions, for example the inversions studies by (Paulot et al., 2014;Zhang et al., 2018). **However, more attention should be paid to the validations of model before the inversion estimation of NH₃ emissions. How to represent the model uncertainties in the current framework of emission inversion is also an important aspect in future studies.** Things could be better for CO considering its small and weakly spatial-dependent model uncertainties."*

**Changes in the revised manuscript: lines 454–461.**

**Other specific comments:**

**Comment 6:** In page 1, line 40, change "peral"to"pearl".

**Reply:** We have revised it.

**Changes in the manuscript: lines 41**

**Comment 7:** In page 4, line 4, missing "plain"

**Reply:** We have revised it.

**Changes in the manuscript: lines 111-112.**

**Comment 8:** In Figure 1, I think the color of CO measurement sites in NCP should be "green" instead of "blue".

**Reply:** We have revised it.

**Changes in the manuscript: lines 776.**

We Thank Reviewer for his/her constructive comments

Responses to the Specific comments

**General comments:** This work evaluated 14 model simulations of NO2, CO and NH3 over China under the framework of MICS-Asia III with the aim to assess the capability and uncertainty of current CTMs in East Asia. Model results were provided by a larger number of independent groups and covered a full year (2010). The results show that most models well captured the monthly and spatial patterns of NO2 in NCP though NO2 levels are slightly underestimated, but relatively poor model performance was observed in the PRD region. All models significantly underpredict CO concentrations both in the NCP and PRD regions and failed to reproduce the observed monthly variation of NH3 in NCP. This work quantifies the impacts of model uncertainties on simulations of the three primary gases, which shows the large uncertainty (spread) in simulating more reactive and/or short-lived primary pollutants (e.g. NH3). This work is important and valuable to the scientific and regulatory community as it provides information on the capability and limitations of some widely used models. The manuscript is well organized and well written, and model results (tables and figures) are clearly presented. I recommend its publication after the authors have addressed my comments listed below.

**Reply:** The authors appreciate the reviewer for his/her valuable suggestions. In the revised manuscript we have considered each comment for improvement, revision, and correction. Please refer to our responses for more details given below.

**Comment 1:** For comparison with the NO2 measured from the regular monitoring networks, please note that these networks employ a thermal conversion method which converts NO2 to NO, followed by detection of NO. This method is known to overestimate NO2 as it also converts other NOy species such as HONO and PAN etc (e.g., Xu et al., 2013). It is important to correct this measurement problem before making the comparison, using, for example, the approach by Zhang et a. (2017). After corrections of the measurement data, a closer agreement would be seen between the modelled results and the observations in the present work. If the author cannot make such corrections in view of a large number of groups involved, at least some discussions should be provided on this point.

**Reply:** Thanks for this important point. According to Xu et al., 2013, the thermal conversion method has a problem of overestimating the NO₂ concentrations due to the positive interference of other oxidized nitrogen compounds. Zhang et al., 2017 has proposed a method to correct this measurement error based on the model simulations using the equation of:

$$NO_{2\,obs} = NO_{2\,obs}^* \times \frac{NO_{2\,mod}}{NO_{2\,mod} + NO_{z\,mod} - Nitrate_{mod}}$$

where $NO_{2\,obs}$ is the corrected NO₂ observations; $NO_{2\,obs}^*$ is original measurement of NO2; $NO_{2\,mod}$ is the simulated NO₂ concentration; $NO_{z\,mod}$ is the sum of simulations of HONO, 2×N₂O₅, ClNO₂, ClONO₂, NO₃ HNO₃, HNO₄, PAN, and Nitrate; and $Nitrate_{mod}$ is the simulated nitrate.

However, as the reviewer mentioned, it is hard to make such corrections using a large number of models due to the model uncertainties in predicting the concentrations of NO₂, NO_Z and Nitrate. Thus, following the suggestions of reviewer, we have added the discussions of the positive biases in the measurement NO₂ concentrations in the revised manuscript (*see lines 190–192 in the revised manuscript*), which as follows:

*"It should be noted that these networks measured the NO₂ concentrations using a thermal conversion method, which would overestimate the NO₂ concentrations due to the positive interference of other oxidized nitrogen compounds (Xu et al., 2013)."*

According to this, the underestimated NO₂ predictions by the models may also be related to the positive biases in the NO₂ observations, which has been clarified in the revised manuscript (*please see lines 234–236 and lines 416–417 in the revised manuscript*).

**Changes in the manuscript: lines 190-192, lines234-236 and lines 416-417.**

**Comment 2:** Section 2.2. The comparison of NO2 and CO concentrations are only for NCP and PRD. Any reasons why not to include other regions?

**Reply:** Thanks for this comment. This manuscript focuses on the evaluation and uncertainty investigation of NO₂, CO and NH₃ modeling over China under the framework of MICS-Asia III. The CTMs were run at the base year of 2010 when the observations were very limited in China, thus observation data for NO₂ and CO concentrations only included that from Chinese Ecosystem Research Network (NCP), Pearl River Delta Regional Air Quality Monitoring Network (PRD RAQMN) and the Acid Deposition Monitoring Network in East Asia (EANET). Since the observation data from EANET was very limited in China, we only evaluated the CO and NO₂ modeling results in the NCP and PRD regions, the two typical industrialized regions in China. In next phase of MICS-Asia (MICS-Asia IV), more observations will be available in China, which would allow us a more thorough evaluation of the model performance over China.

**Comment 3**: For simulations of NO2 (and NH3), accurate representation of nitrogen chemistry is critical. Recent studies have shown that the HONO sources may be under-represented in some models which would give rise to larger simulated NO2 values (as it underestimates the oxidation of NO2 by OH) (e.g., Zhang et al., 2017; Fu et al., 2019); N2O5 uptake on aerosol may be treated differently in models which could also affect the NO2 simulations. Therefore, in discussing the discrepancy in modelled NO2, information on how models treat these nitrogen processes would be helpful.

**Reply:** Thanks for this comment. We agree with the reviewer that the HONO chemistry has an important role in the nitrogen chemistry in the atmosphere, which influences the simulations of $NO_2$ and $NH_3$(Fu et al., 2019;Zhang et al., 2016;Zhang et al., 2017). Previous studies also indicated that the HONO sources were commonly underestimated in models (Zhang et al., 2016). The heterogenous reactions of $NO_2$ on the surfaces ($2NO_{2(g)} + H_2O_{(l)} \rightarrow HONO_{(l)} + HNO_{3(l)}$) was one of the dominant sources of HONO in the atmosphere, which has been considered in most models of MICS-Asia III, including CMAQ since version 4.7, NAQPMS, NHM-Chem and GEOS-Chem. However, some other important sources of HONO may still be underestimated by models in MICS-Asia III. For example, Fu et al., 2019 suggested that the high relative humidity and strong light could enhance the heterogeneous reaction of $NO_2$ , and the photolysis of total nitrate were also important sources of HONO. These sources has not been included in the models of MICS-Asia III, which would lead to the deviations from observations. As the reviewer suggested, different treatment of hydrolysis of $N_2O_5$ would help explain the differences in the modeled $NH_3$ concentrations. The hydrolysis of $N_2O_5$ has not been considered in M7, which would leads to a lower tendency in the prediction of $NO_3^-$ (Chen et al., 2019) and may partly explain the higher $NH_3$ predictions in M7.

Based on these results, we have added the discussions of HONO chemistry in the revised manuscript (*please see lines 441–449 in the revised manuscript*).
**Changes in the manuscript: lines 441-449.**

**Comment 4:** The photo-chemical mechanisms used in this study are CBMZ, CB05, and SAPRC 99, and some of them have an updated version such as CB06 and SPARC 07. These updated mechanisms could give different results on model performance. The author is advised to discuss this point to alert the reader that their conclusion may not be applicable to the newer version of the respective mechanism.

**Reply:** Thanks for this important point. We have clarified this point in the revised manuscript (*please see lines 472–474 in the revised manuscript*), which as follows:

*"The gas chemistry mechanisms used in this study are SAPRC 99, CB05, CBMZ, RACM and RADM2, and some of them have an updated version such as CB06 and SPARC 07. Our conclusions may not be applicable to these newer versions of mechanisms and thus more comparisons studies can be performed to understand the differences in these new mechanisms."*

**Changes in the manuscript: lines 472-474.**

**Comment 5**: The present comparisons focused on yearly and monthly model performance. It would be interesting to show how different models compare during severe pollution episodes. An important application of CTMs in China is to forecast severe episodes based on which emergency source control measures are activated.

**Reply:** We agree. Comparisons of different model performance in severe pollution episodes would be very important for the understanding of the capability of current CTMs and their applications in air quality forecast and emission controls. However, in current phase of MICS-Asia, only monthly modeling results has been provided by different CTMs, which limited the comparisons at the yearly and monthly scale. The model performances in pollution episodes will be investigated in MICS-Asia IV with more observation data and hourly simulation results at severe pollution episodes.

**Comment 6:** The model comparisons were conducted for NO2, CO, and NH3. How about SO2, which is another important primary pollutant? I think the reader would be interested in seeing the model performance for SO2 as well.

**Reply:** Thanks for this suggestion. Our study mainly focused on the model performance of $NO_2$, CO and $NH_3$. The model comparisons of $SO_2$ has been covered in a companion paper (Tan et al., 2019), where both the performance of $SO_2$ and sulfate has been investigated.

**Comment 7**: Conclusion (1) recommends to improve the CO emission inventory which is for year 2010. Does the recent CO emission have similar problem?

**Reply:** Thanks for this important point. Since we only evaluated the CO simulations for year 2010, the direct evaluations of CO emissions for recent years were not available in this study. However, we have added some discussions on the recent CO emissions in the revised manuscript (*please see lines 427–433 in the revised manuscript*), which as follows:

*The underestimations of CO emissions may be alleviated in recent years due to the decreasing trends of the Chinese CO emissions in recent years(Jiang et al., 2017;Zhong et al., 2017;Sun et al., 2018;Muller et al., 2018;Zheng et al., 2018;Zheng et al., 2019). The inversion results of Zheng et al., 2018 also agree well with the regional MEIC (Multi-resolution Emission Inventory for China) inventory for CO emissions in China from 2013 to 2015. However uncertainties still exist in the CO emissions in recent years, according to previous studies, the estimated CO emissions for the whole China for year 2013 ranges from 134–202 Tg/yr (Jiang et al., 2017;Zhong et al., 2017;Sun et al., 2018;Muller et al., 2018;Zheng et al., 2018;Zheng et al., 2019). Zhao et al., 2017 also suggested a -29%–40% undertainty of CO emissions from industrial sector in year 2012.*

**Changes in the manuscript: lines 427-433.**

**Comment 8:** This study reveals a large spread of model simulations for reactive gases. As the exact causes for the difference have not been identified for the individual model, I think it is important to emphasize the need to validate the individual model before using its results to make important policy recommendation.

**Reply:** Thanks for this important point. We have clarified this point in the revised manuscript (*please see lines 462–466 in the revised manuscript*), which as follows:

*"For some highly active and/or short-lived primary pollutants, like $NH_3$, model uncertainty can also take a great part in the forecast uncertainty. Emission uncertainty alone may not be sufficient to explain the forecast uncertainty and may cause underdispersive, and overconfident forecasts. Future studies are needed in how to better represent the model uncertainties in the model predictions to obtain a better forecast skill. **Such model uncertainties also emphasize the need to validate the individual model before***

*using its results to make important policy recommendation."*

**Changes in the manuscript: lines 462-466.**

**Minor Comments:**

Line 40 page1, line 4 page 4, the "Peral" should be "Pearl".

**Reply:** We have revised it.

**Changes in the manuscript: lines 41.**

**References**

[revised manuscript text omitted]

**Supplementary Material**

**Sect. S1 Evaluations of the standard meteorological simulations**

Meteorological simulations have large impacts on the simulations of atmospheric chemistry. The simulated wind speed (u-wind and v-wind), relative humidity (RH) and air temperature (T) from the standard meteorological fields were evaluated against the observations over the NCP and PRD regions. These parameters are all important factors that influences the simulations of $NO_2$, CO and $NH_3$. For example, the wind speed determines the transport of species and the air temperature influences the reaction rates of thermal chemical reactions. The RH and T also influence the thermodynamic equilibrium of gases and aerosols.

Three-hourly meteorological observations from the Integrated Surface Database (ISD) compiled by the National Oceanic and Atmospheric Administration (NOAA), U.S. (Smith et al., 2011) were used in meteorological evaluations with observation sites in the NCP and PRD regions shown in fig S1. Figure S2 shows the averaged time series of the simulated and observed meteorological parameters over the NCP region from January, 2010 to December, 2010. The evaluation statistics, including correlation coefficient (R), mean bias error (MBE) and root of mean square error (RMSE), were summarized in Table S2. It clearly shows that the standard meteorology simulations well captured the main features of the observed meteorological conditions in NCP throughout the year with high correlation coefficient, small biases and low errors for all meteorological parameters. Similar results could be obtained from the evaluations of meteorological conditions over the PRD region (fig. S3). These results suggested that the standard meteorological simulations can well reproduce the meteorological conditions of the NCP and PRD regions.

**Sect. S2 Descriptions of the IASI measurement of $NH_3$ total columns**

The ANNI-NH3-v2.1R-I retrieval product (Van Damme et al., 2017;Van Damme et al., 2018) was used in this study to quantitatively evaluate the modeled monthly variations of $NH_3$ concentrations. It is a reanalysis version of $NH_3$ retrievals from IASI instruments and provides the daily morning (~9:30 am local time) $NH_3$ total columns from year 2008 to 2016. The morning orbit was used since IASI is generally more sensitive to the atmospheric boundary layer at this time due to more favorable thermal conditions, which could provide more information on the $NH_3$ concentrations in the boundary layer where $NH_3$ is emitted. The dataset was produced by Van Damme et al., 2018 based on the conversion of hyperspectral range indices (HRIs) using an Artificial Neural Network(Whitburn et al., 2016). It uses the ERA-interim ECWMF meteorological input data rather than the operationally provided EUMETSAT IASI Level 2 (L2) data used for the standard near-real-time version, which is more coherent in time and suitable for the study of temporal variations. To facilitate comparisons, the $NH_3$ total columns were averaged to monthly data at 45km × 45km MICS-Asia grids.

**Sect. S3 Sensitivity experiments of high-resolution simulation in the PRD region**

To investigate the impacts of horizontal resolution on the simulations of gas concentrations over the PRD region, a full- year run with finer horizontal resolutions has been conducted using the NAQPMS model, which is one of the participating

CTMs in MICS-Asia III. In our experiment, two nested domains with finer horizontal resolutions were added to the original modeling domain of MICS-Asia III, which are shown in Fig. S4. The first domain (D1) is identical to the modeling domain of

MICS-Asia III with horizontal resolution of 45km. The second domain (D2) covers most part of southeast China with horizontal resolution of 15km. The third domain has the finest horizontal resolution (5km) which covers the PRD region and its surrounding areas. The chemical configurations of NAQPMS in each modeling domain were completely identical to those used in MICS-Asia III. Meteorological fields for each modeling domain were simulated by the WRF model version 3.4.1, same as the standard meteorological model in MICS-Asia III. The WRF configurations were also kept same as those used in the standard meteorological simulations except two additional nested domains were added (Fig. S4). The emission inventories and boundary conditions in D1 were provided by the standard input datasets of MICS-Asia III. Since MICS-Asia III only provided the 45km-resolution emission inventories and boundary conditions, the emission rates ($\mu g/m^2/s$) and boundary conditions over one model grid in D2 and D3 were simply obtained from the corresponding model grid in its parent domain.

This means that although we used the finer horizontal resolutions in D2 and D3, the resolutions of emission inventories and boundary conditions in D2 and D3 were the same as those used in D1. Therefore, the horizontal resolutions were only dynamically increased in D2 and D3. The modeling results from different modeling domains were then compared with each other to investigate the dynamical impacts of horizontal resolution on the model performance.

Figure S5 shows the spatial distributions of the observed annual mean $NO_2$ concentrations in PRD region overlay the simulated concentrations with different horizontal resolutions. We can clearly see that the coarse modeling results (D1) cannot resolve the high spatial variability of $NO_2$ concentrations in the PRD region. For simulations using finer horizontal resolutions (D2 and D3), although the spatial scales of $NO_2$ observations can be resolved by the 15km and 5km resolutions, the modeling results still show poor performance in capturing the observed spatial variability of $NO_2$ concentrations, with calculated correlation coefficient only of 0.03 and 0.02, respectively (Table S3), even worse than the coarse modeling results. Similar results could be obtained from the comparisons of CO observations and simulations using different horizontal resolutions (Fig.

S6). These results indicated that the poor model performance in PRD may not be attributed to the resolution of model but more related to the resolution and/or spatial allocation of emission inventories in the PRD region. These results also suggested that only increasing the resolution of model may not help improve the model performance.

**Tables:**

Table S1 Configurations of the standard meteorological model and different WRF-Chem models

| No | Microphysics | Longwave radiation | Shortwave radiation | Boundary layer | Cumulus physics | surface physics |
|---|---|---|---|---|---|---|
| Standard | Lin et al. scheme | RRTMG scheme | Goddard shortwave scheme | YSU scheme | Grell 3D ensemble scheme | Unified Noah land-surface model |
| M7 | Lin et al. scheme | RRTM scheme | Goddard shortwave | YSU scheme | Grell 3D ensemble scheme | Unified Noah land-surface model |
| M8 | Lin et al. scheme | RRTMG scheme | RRTMG scheme | Mellor-Yamada-Janjic TKE scheme | Grell 3D ensemble scheme | Unified Noah land-surface model |
| M9 | Lin et al. scheme | RRTMG scheme | RRTMG scheme | YSU scheme | Grell 3D ensemble scheme | Unified Noah land-surface model |
| M10 | Goddard Cumulus Ensemble | Goddard longwave scheme | Goddard shortwave scheme | YSU scheme | Grell 3D ensemble scheme | Unified Noah land-surface model |

Table S2 Evaluation metrics of the standard meteorological simulation

| | NCP | | | PRD | | |
|---|---|---|---|---|---|---|
| | R | MBE | RMSE | R | MBE | RMSE |
| temp (℃) | 1.00 | 0.21 | 1.08 | 1.00 | -0.22 | 0.71 |
| RH (%) | 0.97 | -0.16 | 5.15 | 0.97 | 3.42 | 4.82 |
| u-wind (m/s) | 0.91 | -0.08 | 0.63 | 0.82 | -0.20 | 0.53 |
| v-wind (m/s) | 0.93 | 0.33 | 0.76 | 0.93 | 0.05 | 0.81 |

Table S3: Evaluation metrics of the simulated annual mean $NO_2$ and CO concentrations over the PRD region with different horizontal resolutions.

| | $NO_2$ (ppbv) | | | | CO (ppmv) | | | |
|---|---|---|---|---|---|---|---|---|
| | Spatial R | MBE | NMB (%) | RMSE | Spatial R | MBE | MBE (%) | RMSE |
| 45km | 0.09 | 2.99 | 13.37 | 10.53 | 0.00 | -0.51 | -52.85 | 0.57 |
| 15km | 0.03 | 2.19 | 9.81 | 10.15 | 0.00 | -0.54 | -56.25 | 0.60 |
| 5km | 0.02 | 0.58 | 2.59 | 10.23 | -0.10 | -0.58 | -59.23 | 0.62 |

**Figures:**

[Figure]

**Figure S1: spatial distributions of the meteorological observation sites from the ISD over the NCP region (left panel) and the PRD**

**region (right panel).**

[Figure]

**Figure S2: Time series of the simulated and observed meteorological parameters over the NCP region form January 2010 to**

**December 2010 with an interval of three hours.**

[Figure]

**Figure S3: Same as Figure S\* but for the PRD region.**

[Figure]

**Figure S4: Modeling domain of the sensitivity experiment using different horizontal resolutions. The first domain (D1) is identical**
**to the modeling domain of MICS-Asia III with horizontal resolution of 45km. The second domain (D2) covers most part of southeast**
**China with horizontal resolution of 15km, and the third domain has the finest horizontal resolution (5km) covering the PRD region**
**and its surrounding areas.**

[Figure]

**Figure S5: Spatial distributions of the observed and multi-resolution simulated annual mean NO₂ concentrations over the PRD**

**region.**

[Figure]

**Figure S6: Same as fig.S6 but for CO concentrations.**

[Figure]

**Figure S7: Time series of the surface NH₃ concentrations (left panel) from AMoN-China and NH₃ total columns from IASI (right panel) over the NCP region during September 2015 – August 2016. Note that we reordered the months to better characterise the monthly variations.**

[Figure]

**Figure S8: Monthly series of IASI measured NH₃ total columns over the NCP region from year 2008 to 2016**